# How Neural is a Neural Foundation Model?

## Abstract

Foundation models have shown remarkable success in fitting biological visual systems; however, their black-box nature inherently limits their utility for understanding brain function. Here, we peek inside a SOTA foundation model of neural activity (Wang et al., 2025) as a physiologist might, characterizing each 'neuron' based on its temporal response properties to parametric stimuli. We analyze how different stimuli are represented in neural activity space by building *decoding manifolds*, and we analyze how different neurons are represented in stimulus-response space by building *neural encoding manifolds*. We find that the different processing stages of the model (i.e., the feedforward *encoder*, *recurrent*, and *readout* modules) each exhibit qualitatively different representational structures in these manifolds. The *recurrent* module shows a jump in capabilities over the *encoder* module by "pushing apart" the representations of different temporal stimulus patterns. Our "tubularity" metric quantifies this stimulus-dependent development of neural activity as biologically plausible. The *readout* module achieves high fidelity by using numerous specialized feature maps rather than biologically plausible mechanisms. Overall, this study provides a window into the inner workings of a prominent neural foundation model, gaining insights into the biological relevance of its internals through the novel analysis of its neurons' joint temporal response patterns. Our findings suggest design changes that could bring neural foundation models into closer alignment with biological systems: introducing recurrence in early encoder stages, and constraining features in the readout module.

## 1 Introduction

Viewed in the large, deep neural networks are intriguing models of the mouse visual system, since they learn to predict neural responses directly from visual input (Cowley et al., 2023; Ustyuzhaninov et al., 2022; Huang et al., 2023; Averbeck et al., 2006; Qazi et al., 2025; Li et al., 2023), and recent foundation models can generalize, to some extent, beyond training data (Li et al., 2023). Viewed in the small, Representational Similarity Analysis (RSA) Kriegeskorte et al. (2008) shows that, on average, many units in these networks reflect properties (e.g. orientation selectivity) resembling those found in biology (Conwell et al., 2021; Qazi et al., 2025). However, while this progress has been impressive, questions are arising about whether the pairwise activity of units in artificial networks agrees with biological data (Liscai et al., 2025). Moreover, in the large view the input/output maps are far from complete (normalized response correlation ceilings around 70% (Wang et al., 2025)), raising questions about their robustness. In effect, response correlation measures how well the input drives the system to the correct output; it does not address the inverse question of how ambiguity in the output obscures the input. That is, one must consider both the "forward" and the "inverse" mappings. Such issues are classical in modeling: control theory teaches us that, without a perfect model, one must "look inside the box" to achieve identifiability (cf. (Åström, 2012)). We seek to do just this on the Foundation Neural Network (FNN) (Wang et al., 2025). Without this, we cannot guarantee correct, robust, and generalizable behavior, especially on out-of-distribution data, to confidently build hypotheses about the brain using the FNN. The FNN was selected because it was trained on MICrONS, the largest available functional connectomics dataset of the mouse visual system (Bae et al., 2025) and is based on artificial and naturalistic input videos across multiple animals. The FNN thus provides the SOTA in modeling.

FNN consists of multiple stages (Figs. 1B and 6) and millions of units, so analyses beyond pairwise interactions—such as third- or fourth-order statistics—are computationally prohibitive. To "look inside," we use three techniques popular in neuroscience. These allow us to: (1) evaluate how the state of the network represents the different stimuli; i.e. how stimuli are related to one another in global

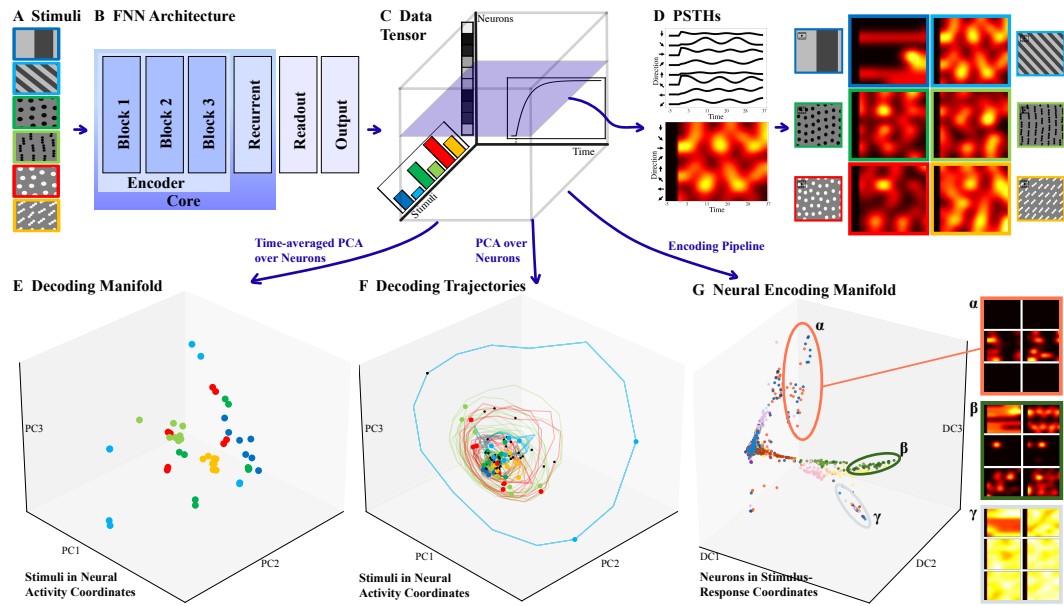

Figure 1: **Approach and manifolds analysis A** Stimulus ensemble provides input. **B** FNN consists of multiple encoding blocks, modeled as convolutional layers, followed by recurrent and read-out/interpolation layers. **C** The tensor of data, containing the response (in time) of each sampled unit to the stimulus ensemble. **D** PeriStimulus Time Histogram: The response (instantaneous "firing rate") of a single unit/neuron to a stimulus pattern drifting in each of 8 different directions. The curves are redrawn as an image, with brightness corresponding to activity. A plane through the data tensor shows the PSTHs for each of the 6 stimulus classes, drifting in all directions. **E** Decoding manifold, plots the total activity for each stimulus in PCA-reduced neural coordinates. Colors correspond to stimulus classes in **A**. **F** The time evolution of each stimulus presentation, plotted in PCA-reduced neural coordinates for the early encoder layer. Note the nested, periodic trajectories indicating a stimulus drifting over a receptive field filter. **G** Encoding manifold plots individual units/neurons in stimulus/response coordinates. Note the clustering of units with similar responses across the ensemble.

neural coordinates (Fig. 1E); (2) to show how all the units in the network are related to each other functionally when driven by the stimuli (Fig. 1G); i.e. how they encode information; and (3) how the dynamics evolve as the network processes the different stimuli; i.e., how the global neural state changes in time during a computation (Fig. 1F). The first two techniques result in manifolds characterizing *forward* and *backward* mappings, respectively, and the third in trajectories over these manifolds; all can then be compared against biology. The result, in brief, is that while the FNN learned a forward map reasonably well, it processes stimuli quite differently from the mouse, and hence is only a partial "digital twin" in the dynamical sense. Importantly, our manifolds identify where the disparities lie.

In more detail, (1) we build *neural decoding manifolds* (Chung and Abbott, 2021), in which trials are embedded in the space of neural activity coordinates (Fig. 1E), then dimensionality-reduced using Principal Component Analysis (PCA) (Cunningham and Yu, 2014). Typically, trials involving the same stimulus cluster together, facilitating a read-out of the brain's state. (2) To switch from trials to neurons, we build *neural encoding manifolds* (Fig. 1G) (Dyballa et al., 2024a) in which each point is a neuron in the space of stimulus-response coordinates, dimensionality-reduced using tensor factorization (Williams et al., 2018). Proximity between neurons in an encoding manifold denotes similar responses to similar stimuli; i.e., groupings of neurons that are likely to share circuit properties. For a review of classic encoding/decoding in neuroscience, see (Mathis et al., 2024). Finally, (3) the relationship between these two manifolds is captured by the temporal evolution of each neuron's activity for each stimulus trial. Recalling that a 'neural computation' can be viewed as the result of a dynamical system in neural state space (Hopfield, 1984), we plot these both as PeriStimulus Time Histograms (PSTHs, Fig. 1D) and as streamline traces (decoding trajectories, Fig. 1F). While streamline representations have been used previously for decision tasks (Duncker and Sahani, 2021) and the motor system (Churchland et al., 2012; Safaie et al., 2023), we note: (i) the activity integral along such *decoding tra-*

*jectories* (Fig. 1F) defines the decoding manifold, while (ii) shared tubular neighborhoods (developed below) specify position in the encoding manifold. These three perspectives enable us to investigate different aspects of alignment: (1) Decoding manifolds reveal whether the model maintains stimulus separability like biology; (2) Encoding manifolds reveal whether functional topology of neurons is brain-like; (3) Trajectories reveal whether the model performs computations through brain-like dynamics. Critically, a model could succeed at one level while failing at others. We use modeling tools available online (references in Methods), stimuli similar to those used in FNN's original training (Wang et al., 2025), and add naturalistic flow stimuli used in mouse physiology (Dyballa et al., 2018) (Fig. 1A).

**Prior Work**. There is an extensive literature on modeling biological neural responses (Averbeck et al., 2006; Ustyuzhaninov et al., 2022; Qazi et al., 2025), including other foundation models (Zhang et al., 2025; Azabou et al., 2023; Ryoo et al., 2025; Ye et al., 2023, 2025). We highlight that compared to these other approaches, the FNN is concerned with predicting neural activity from input videos. The FNN is an example of a data-driven predictive model (Klindt et al., 2018; Turishcheva et al., 2024; Nellen et al., 2025) with Gaussian readout (Lurz et al., 2021) that interprets the readout as per-neuron basis functions with individual readout weights. The readout thus provides an encoding embedding of biological neurons. For comparability, we use our encoding method to compare the embeddings of biological neurons and individual readout neurons, investigating not only the final embedding but also the readout embedding. Different loss functions have been used (Nayebi et al., 2023; Bakhtiari et al., 2021; Shi et al., 2022), and others have studied decoding manifolds (Froudarakis et al., 2020; Beshkov and Tiesinga, 2022; Beshkov et al., 2024), focusing on topological properties. For a recent general review, see Doerig et al. (2023). Some studies are supportive of modeling brains with deep networks (Kriegeskorte, 2015; Yamins et al., 2014; Margalit et al., 2024), while others raise questions (Serre, 2019a). For the reasons stated above we focus on the FNN.

To our knowledge, this is the first time all three of the encoding and decoding manifold techniques have been utilized together for analysis of a perceptual system; i.e., toward *interpretability* for a foundation model. Interpretability is a rapidly evolving field for analyzing large language models (Elhage et al., 2021; Bricken et al., 2023; Skean et al., 2025), vision models (Simonyan et al., 2014; Olah et al., 2017), and recurrent models (Krakovna and Doshi-Velez, 2016). This field has been connected to neuroscience, arguing that both aim to understand complex intelligent black boxes (Kar et al., 2022; Tolooshams et al., 2025; He et al., 2024; Mineault et al., 2025). It aims to investigate the function of individual neurons, circuits, and modules in artificial networks, while in neuroscience it additionally focuses on the alignment between artificial models and biological systems (Kar et al., 2022). We tackle both challenges by trying to understand what functions the FNN modules fulfill and by testing alignment with biological representations.

Within this framework, we ask: *Do neural decoding and encoding manifolds reveal new insights into how foundation models represent temporal response patterns? Are their representations brain-like?* We hypothesize that each processing stage contributes distinct representational capabilities, all essential for fitting neural data. In particular, one might expect the *recurrent* module to enrich the temporal structure of representations, analogously to the cortex, and the encoder layers to resemble the retina with its limited recurrence. Following a brief description of our methods, we proceed to develop each of the manifolds in turn.

## 2 METHODS

Our work makes novel use of publicly available open-source resources. Specifically, we employed the pretrained foundation model of neural activity (denoted FNN) provided by Wang et al. (2025), available here; and the stimulus generation tools and neural encoding manifold construction pipeline introduced by Dyballa et al. (2024a), accessible at here. Below we briefly outline our methods, and refer readers to Appendix A for the full details.

**Model:** The FNN consists of five modules: perspective, modulation, *encoder*, *recurrent*, and *readout* (see Fig. 6). The perspective and modulation modules model the mouse's state and transform the inputs to approximate the actual visual information received. Thus, only the *encoder*, *recurrent*, and *readout* modules perform the core computation, and are the focus of this work. The *encoder* module is a 10-layer DenseNet-style convolutional encoder (Huang et al., 2017). For analysis, we use a subset of *encoder* layers; we report results from the very first layer and the last block as representative examples (remaining layers in the appendix). Notably, the encoder includes 3D convolutions, which

in principle enable it to capture temporal patterns for up to 12 timesteps in the last encoder layers. The *recurrent* module is preceded by an attention layer and consists of a convolutional LSTM, followed by a single convolutional layer that produces its output. This feedforward–recurrent combination constitutes the core of the FNN, which is trained on data from all mice combined. Finally, a separate *readout* module is trained on each mouse individually: it performs an interpolation on the recurrent output followed by a linear transformation to produce the FNN output. We included one scan (session 8, scan 5) for readout and output analysis, and validated the findings on other sessions and scans. We claim that comparison across mice on the *population* level, rather than on individual neurons, is valid.

**Stimuli:** Our stimulus set is composed of drifting square-wave gratings and optical flows with varying spatial frequencies moving in eight directions. This yields 88 unique input sequences with stochastic initial positions and velocities (Fig. 1A). To ensure that these stimuli would drive the network in a representative manner, we compared the output of the network for these stimuli with the output for the original natural movie stimuli used to train the network (Appendix Figs. 9 and 10); we found the results to be quantitatively similar in all measured respects.

**PSTH visualization:** To visualize the network responses to stimuli concisely, we group together the model's PeriStimulus Time Histogram responses (PSTH) corresponding to all flow directions of a given stimulus pattern with time on the $x$-axis and flow direction on the $y$-axis (Fig. 1D). **Decoding manifolds & trajectories:** Following traditional analysis techniques, we first constructed decoding manifolds by performing PCA on the stimulus-time-averaged activity data. Therefore, the decoding manifold contains 48 points, one for each unique sequence, colored by the corresponding base-stimulus (as shown in Fig. 1A); different spatial frequencies of the same stimulus are summarized with the same color. To construct *decoding trajectories*, we treated each time step as a separate data point rather than averaging across time before applying PCA. We compared with biological decoding trajectories using the experimental data from Dyballa et al. (2024a).

**Tubularity:** To investigate neural dynamics, we modeled trajectories by bundling them into tubular neighborhoods around a central skeleton (Budanur, 2023). We operationlized this idea for discrete data using the tubular neighborhood theorem (Da Silva, 2008), which guarantees that smooth submanifolds admit non-intersecting neighborhoods diffeomorphic to their normal bundles. Let $\{\gamma_i\}_{i=1}^m \subset \mathbb{R}^D$ denote a set of $m$ trajectories (curves). We define this set as *tubular* if it remains close to a common centerline $c$ and exhibits minimal transverse intersections. Formally, the tube is obtained by expanding $c$ with a radius profile $R(\cdot)$ such that all points at parameter $u$ within distance $R(u)$ of $c(u)$ are included. In practice, curves are first clustered (e.g., via HDBSCAN (Campello et al., 2013) using the Sobolev $H^1$ metric, or with ground truth) to separate distinct tubes before computing tubularity scores. We introduce *tightness*, which measures how tight a group of curves is around the centerline, and *crossings*, measuring how many transverse crossings occur in each trajectory bundle. Therefore, tubularity is not a trajectory-matching metric but a population-geometry metric: it assesses the structure of collections of trajectories rather than the similarity of individual pairs.

**Alignment metrics:** To validate our results against the literature, we calculated scores for Representational Similarity Analysis (RSA) (Kriegeskorte et al., 2008), Canonical Correlation Analysis (CCA) (Raghu et al., 2017), Linear Predictivity (LP) (Yamins et al., 2014), and Dynamic Similarity Analysis (DSA) (Ostrow et al., 2023) (details in Appendix A.12)

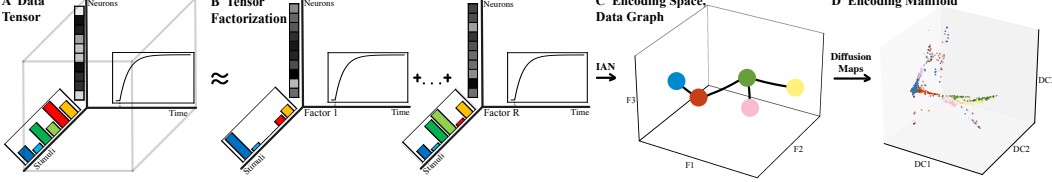

Figure 2: **Encoding Manifold Pipeline (A, B)** A non-negative tensor factorization of the original data tensor identifies those neural factors that account for part of the stimulus ensemble over comparable time epochs. **(C)** Collecting the neural factors into a linear vector space, an adaptive-neighborhood kernel builds a data graph. **(D)** Diffusion maps yield the encoding manifold.

**Encoding manifolds:** To understand the response properties of *neurons* with respect to all stimuli (rather than the representation of *stimuli* in the space of all neurons), we finally constructed *encoding manifolds*. At a high level (Fig. 2), these manifolds allow one to examine the global topology of neuronal populations based on their stimulus selectivities and temporal response patterns (Dyballa

et al., 2024a). The neural encoding manifold was constructed in three steps. First, a 3-tensor was built with the temporal responses from each neuron for each stimulus, and decomposed using Nonnegative Tensor Factorization (details in Appendix), with each component comprised of neural, stimulus, and temporal response factors. The neural factors then serve as position coordinates, embedding the neurons into a stimulus-response framework called the neural encoding space. Second, we constructed a data graph in this neural encoding space using the IAN algorithm (Dyballa and Zucker, 2023). Third, applying diffusion maps (Coifman et al., 2005; Coifman and Lafon, 2006) to the data graph yielded the manifold. We followed the methodological choices of Dyballa et al. (2024a), where extensive parameter analysis for biological neural data was conducted.

# 3 RESULTS

We built encoding and decoding manifolds, as well as decoding trajectories, for all layers of the modules considered in the FNN. Here, we focus on the results that were most informative for interpreting the computational role of each stage of the network and for comparing the FNN representations to biological results (see Appendix for extended results). The **decoding manifolds** assess *stimulus separability*, the **encoding manifolds** capture *global neuronal response similarity and topology*, and the **trajectories** characterize *response dynamics*. Together, these analyses provide complementary perspectives for evaluating brain alignment at the population level.

## 3.1 DECODING MANIFOLDS

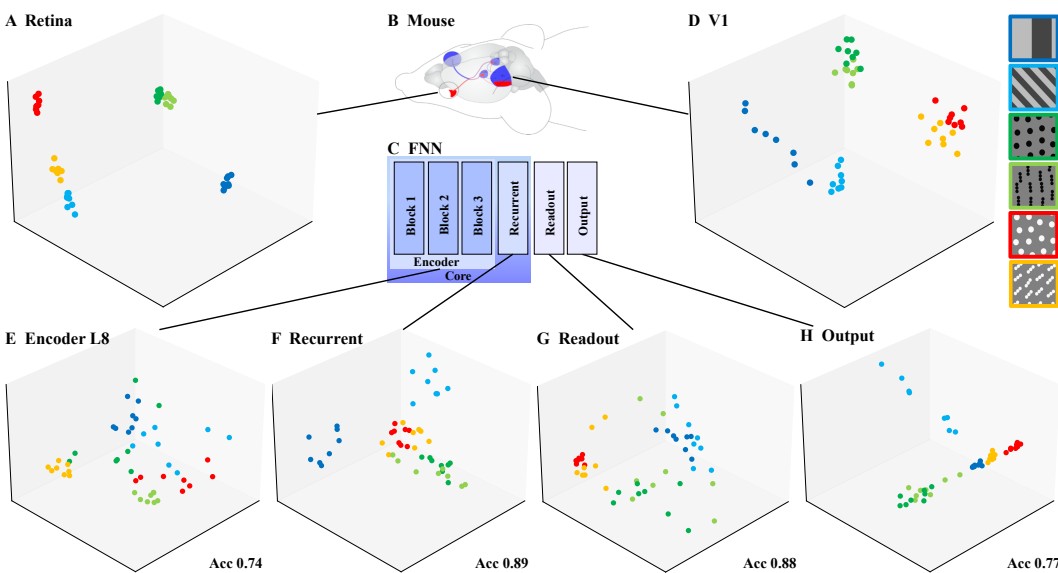

Figure 3: **Decoding Manifolds** for the mouse (**A**) retina and (**D**) visual cortex are highly clustered by stimulus (color labels shown in top-right bar) supporting decoding (i.e., reading out the stimulus from neural responses) in both cases. By contrast, the FNN is most clustered at the recurrent and readout stages (**E–H**). Acc: classification accuracy for that layer (see Table 1). Notice how the encoder (first stage in the FNN) differs significantly from the retina (first stage in the visual system); on the other hand, the recurrent layer is most analogous to V1.

Table 1: Stimulus classification accuracy for Leave-One-Out 3-Nearest Neighbor (3-NN) and Logistic Regression (LR) classifiers trained on each layer's activations. Methods in Appendix A.

| Accuracy | L1 | L2 | L4 | L5 | L7 | L8 | Rec | RecOut | Readout | Out |
|---|---|---|---|---|---|---|---|---|---|---|
| LR | 0.59 | 0.62 | 0.66 | 0.65 | 0.71 | 0.74 | 0.89 | **0.90** | 0.88 | 0.77 |
| 3-NN | 0.41 | 0.66 | 0.58 | 0.52 | 0.53 | 0.61 | **0.73** | 0.64 | 0.63 | 0.67 |

The biological decoding manifolds (Fig. 3A, D) showed clear clustering by stimulus with some overlap between the related 1-dot and 3-dot stimuli. It follows that neural responses at both the retina and cortical levels can be used to "read out" the stimulus. By contrast, the first encoder layer (L1) yielded a poorly clustered decoding manifold (Fig. 1E) in which stimulus classes were mixed. This implies that the latent feature representation at this point within the FNN is not sufficient to distinguish between the different stimuli (indeed, its classification accuracy is lowest; see Table 1). The decoding manifold for layer 8 (L8) was similar to that for L1, but with greater stimulus-specific clustering. **The recurrent decoding manifold was closest to the biological data, showing more distinct clusters and greater overlap between 1-dot and 3-dot stimuli.** Following this, the readout and output decoding manifolds showed weaker clustering, suggesting these stages are responsible for fitting neural data rather than enriching the model's representations. This aligns with **the classification accuracy being highest for the recurrent stage and dropping again afterwards, rather differently from biology.**

## 3.2 ENCODING MANIFOLDS

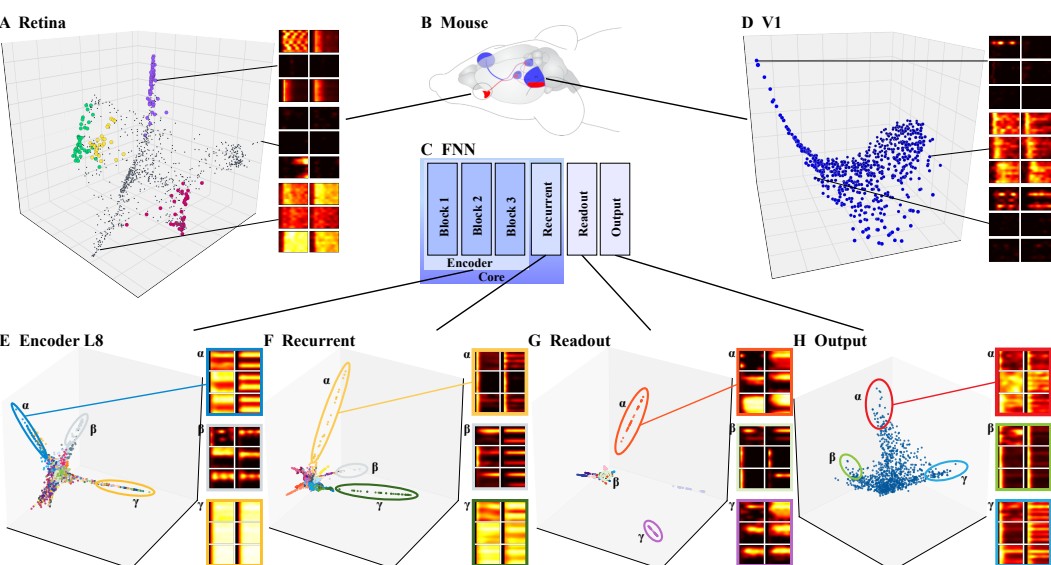

Figure 4: **Encoding Manifolds** for the mouse **(A)** retina and **(D)** visual cortex differ significantly: retina is clustered and cortex is continuous. Example PSTHs show how functionality varies smoothly in cortex but not in the retina. **(E)** The encoder stage showed a distinct arm of orientation-selective units ($\alpha$), which are compatible with biological results, and another of intensity-based units ($\gamma$), which are not. **(F)** The recurrent stage showed many direction-selective units, but the following **(G)** readout stage was the most clustered among all stages. This "bottleneck" layer is then interpolated to a continuous **(H)** output layer. While the topology of this final layer is similar to that of biological visual cortex, the responses of individual units (PSTHs) are not.

The encoding manifolds were even more revealing about differences between the mouse and FNN. Replotting data from Dyballa et al. (2024a), we start with the retinal manifold (Fig. 4A). The neurons form clear clusters, each one with distinct response patterns (PSTHs) that corresponded to known retinal ganglion cell types. By contrast, the V1 encoding manifold is continuous, with smooth transitions in response patterns as it is traversed. See Dyballa et al. (2024a) for further discussion.

The encoding manifold for L1 (Fig. 1G) revealed that most neurons belonging to the same feature map (points with the same color label) formed contiguous clusters, or regions, over the manifold; this was not entirely surprising given the weight-sharing property of these convolutional layers. Nevertheless, several feature maps were found mixed into the same "arm" (labeled $\beta$). Examining the response patterns (PSTHs) of these neurons in detail, we observed strong, continuous activity across the entire trial duration with no selectivity for directions or stimulus classes. There was no biological counterpart to this type of neurons.

We now move on to the late-stage encoder layer, L8 (Fig. 4 E). Its encoding manifold again showed grouping by FNN feature maps, but with more mixing than in L1. This was especially true in the

poorly selective "intensity arm" of neurons, ($\beta$) which exhibited strong response (PSTHs) for all stimuli across multiple feature maps. Further investigation revealed that the intensity arm resulted from an FNN technical requirement: padding artifacts at the edges of feature maps. Such artifacts are a well-known issue in convolutional models (Alsallakh et al., 2020), and we also observed them in Du et al. (2025)'s model (Fig. 17). Sampling only from the central regions of feature maps eliminated both the intensity arm and the shared activity pattern seen in the decoding trajectories (see Supplemental Fig. 16). Although these artifacts distort the representation—indeed, the smoothness of the intensity arm reflects how padding-related information propagates across feature maps—they are part of the network's normal operation. Excluding them would therefore misrepresent the model's true internal dynamics, so we retained them in our manifold analysis.

We emphasize that the non-selective groups of neurons with high activity (labeled as $\beta$ in Figs. 1E and 4E) were a significant departure from what is found in biological networks: in the retina, there are no such non-selective neurons. Although low selectivity has been observed in cortex, it is restricted to inhibitory (inter)neurons and continuously mixes with other, more selective responses; they do not segregate into an arm or cluster (Dyballa et al., 2024a).

The *recurrent* module was qualitatively different. Its encoding manifold showed that different regions exhibited distinct selectivity and temporal response patterns, as evidenced by their PSTHs (Fig. 4F). Furthermore, although segregation by feature map was still present, there was no longer a cluster of neurons with no selectivity; instead, the highlighted $\beta$ group showed selectivity for particular directions or orientations, as is typical in biological visual neurons (e.g., PSTHs in Fig. 4D).

The final stages of the network—the *readout* and *output* layers—were again different. The encoding manifold for the readout layer analyzes the intermediate readout neurons in stimulus-response space, not the final biological output neurons. **It was highly disconnected** (Fig. 4G), with each cluster corresponding almost exclusively to neurons sampled from a single feature map. Each feature map exhibited a distinct response pattern that was invariant across its neurons. Compared to this, the biological results (e.g., Baden et al. (2016); Dyballa et al. (2024a)) showed more variability within functional cell "types", even in the retina. Curiously, and despite this intra-map uniformity, the large number of feature maps (see PSTHs) and the rich dynamics within each one, somehow enable the *output* to represent the complex behavior of neurons (Fig. 4H). These behaviors are captured in the FNN output via a linear combination of *readout* features. Since classification accuracy has declined slightly at this stage (Supplemental Fig. 8), but orientation and direction selectivity agree (Supplemental Fig. 10), we conjecture that these dynamics interpolate the spiking activity individually for each mouse data used as input. **The smooth manifold aligned most closely with the biological V1 manifold** (Fig. 4D), although **the large number of transient responses in the FNN did not match what was found in V1** (across different animals, scans, and sampling procedures).

### 3.3 Decoding trajectories

The encoding manifolds revealed functional differences between FNN and biology: both in the topology of the neuronal organization, and in the PSTHs i.e. temporal responses for multiple stimulus classes. This motivated a direct analysis of the population response dynamics. The **biological decoding trajectories showed stimulus-dependent development of activity** (Fig. 5A,D). They formed segregated, stimulus-dependent bundles whose temporal dynamics allowed linear separability during much of the trial's time course. Here, V1 activity showed more bundles and less collinear development of trajectories. This indicates a higher complexity of response patterns in V1 compared to the retina.

Turning to FNN, the decoding trajectories for L1 revealed that periodic stimuli were represented as loops (Fig. 1F). This was likely due to the translation equivariance of the convolutional layers used in the encoder stage, which preserved the circular geometric structure of these stimulus sequences (Cohen and Welling, 2016). However, we saw that these loops could take on many different forms (such as that for the high spatial frequency gratings, shown in light blue), influenced by the responses of particular groups of neurons to each stimulus. Layer 8, by contrast, showed stimulus-independent temporal decoding trajectories (Fig. 5E). Our analysis of removing the intensity arm from the encoding manifold showed that this temporal development of activity could be attributed to an non-selective increase in intensity during the first timesteps (Supplemental Fig. 16). Without the intensity arm, L8 has highly stationary neural activity. Thus, despite temporal convolutions, the **FNN feedforward encoder appears to lack biologically plausible stimulus-dependent temporal patterns** and primarily reports features present in the input, with varying intensities.

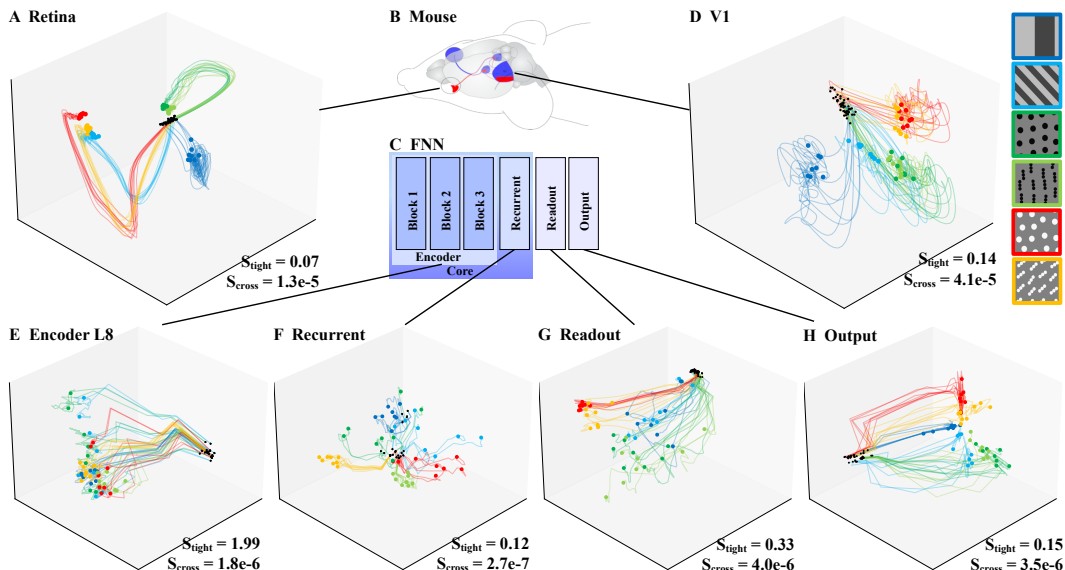

Figure 5: **Decoding Trajectories** in the retina (**A**) and V1 (**D**) show the development of neural activity dynamics into stimulus tubes. The encoder (**E**) shows only a non-selective increase in activity (see also Figure 16) rather than stimulus-dependent tubes. From the recurrent stage onward (**F–H**), tubular trajectories similar to those seen in biological data are present. The tubularity metrics quantify this phenomenon ($S_{tight}$), and also highlight a lack of complexity in FNN activity compared to the biological data, reflected in their lower crossings values ($S_{cross}$).

The recurrent module showed a qualitative change in decoding trajectories compared to the encoder (Fig. 5F). **Similarly to the biological results, tubular temporal patterns were present at the *recurrent* stage**. Still, the organization of decoding trajectories was noticeably more entangled than both retina and V1 (compare with Fig. 5A,D). This phenomenon was quantified using *tubularity* metrics based on the geometry of the observed decoding bundles (see Methods). The tightness scores were comparable between biological and FNN data from the recurrent stage onward (Fig. 5, Table 4). The retinal trajectories were the tightest, while V1 and FNN trajectories showed more expanded cones of trajectories. In particular, the FNN readout trajectories were less tight because they linearly spread out from the origin. The recurrent trajectories were also spread out, but retained a tight stimulus-dependent organization towards the end of the time frame. The tightness score for trajectories from the output stage was difficult to interpret: the predominance of transient responses caused a convergence towards a common point of low activity, which might bias the tightness score to be lower.

A more pronounced difference was observed in the **crossings scores**. Biological trajectories exhibited more crossings than those of the FNN, despite their tight tubular development ($p < 0.005$, Bonferroni-corrected, for all layers). These crossings occurred predominantly toward the end of the time frame, when the activity seemed to settle into a steady state. Several factors could explain this pattern. One possibility is that biological recordings contain inherently more noise, which could artificially lead to more crossings. However, the noise observed toward the end of the biological trajectories is of similar magnitude as the overall tube diameter. If measurement noise were the only cause, we would expect less coherent (tubular), more erratic (noisier) trajectory development already at earlier time steps, which is not observed. A second possibility is that the crossings reflect genuine neural dynamics captured in the data, suggesting that biological systems exhibit more complex temporal processing than the FNN. Modulatory phenomena such as clique-like interactions (Miller and Zucker, 1999) or traveling wave activity (Pitts and McCulloch, 1947; Milner, 1974; Keller et al., 2024) could generate these apparent fluctuations late in the trial. These results indicate that while parts of the FNN reproduce certain aspects of biological temporal structure (such as tubular structure), it is not yet capable of fitting the full intricacies of neural dynamics observed in real neural populations.

The readout and output stages exhibited tubular trajectories that were less well separated than those observed in retina and V1 (Fig. 5G,H), consistent with the less clustered organization seen in the decoding manifolds. In the output trajectories, the bias towards transient responses was clearly visible

as all trajectories originated from a common point (black, high activity) and converged toward a shared low-activity point via different paths.

### 3.4 Representational alignment metrics

To validate the results of our manifold analysis, we quantified the representational alignment of the FNN with both V1 and retina using standard alignment metrics from the literature (Kriegeskorte et al., 2008; Raghu et al., 2017; Yamins et al., 2014; Ostrow et al., 2023). We found that our result of the recurrent module being most aligned with biology in terms of decoding analysis was supported by these metrics (see Tables 2, 6). The DSA metric (Ostrow et al., 2023), while correctly showing higher values for tubular dynamics in the recurrent stage and after, wrongly predicted high alignment between the FNN's L1 and the biological data. This is likely due to tubular trajectories arising for entirely different reasons (i.e., local stimulus periodicity). Moreover, **smoothness and neuronal responses (PSTHs) in the encoding manifold showed a clear misalignment between the FNN's recurrent stage and V1. This relationship was not captured by the standard metrics**, underscoring the need for our analysis at the population level.

## 4 Discussion

**Decoding manifolds and trajectories** allow us to assess whether networks achieve comparable degrees of stimulus representation and separability. **Encoding manifolds**, on the other hand, evaluate at a global level how the responses and global organization of individual neurons compare to those in biological systems; in other words, whether the FNN and biological networks employ similar encoding mechanisms to produce similar outputs. Finally, **decoding trajectories** serve as a surrogate for *computation*, reflecting the dynamics of activity over the neural state space (cf. (Hopfield, 1984)). Our analysis of the FNN revealed an increasing richness of representation up to the *recurrent* module (cf. Hoeller et al. (2024); see also contrasts with Xu et al. (2023); Nayebi et al. (2023); Froudarakis et al. (2020)). However, most PSTHs lacked the characteristic temporal response profiles observed in biological recordings Ringach et al. (2016); Ko et al. (2011). Since the FNN was trained to predict neural spike trains, **classification behavior evolved implicitly** (cf. Table 1)). Thus, it is plausible that the recurrent features are sufficiently complex for robust feature representation and that the subsequent modules serve to fit the neural data rather than to provide additional biologically meaningful computations.

However, the highly clustered topology of the latent representation observed in the *readout* module was not consistent with that of the retina or cortex (cf. Baden et al. (2016); Dyballa et al. (2024a), nor with those of higher visual areas (cf. Glickfeld and Olsen (2017); Dyballa et al. (2024b); Yu et al. (2022)). Nevertheless, the rich dynamics within each feature map (as evident in the PSTHs), together with their large number, seem to enable the output layer to capture the complex response patterns of neurons, resulting in the network's strong performance in predicting neural activity. Still, it is somewhat surprising that such biologically realistic outputs are produced at the FNN's output through a simple linear combination of readout features—one would expect the fitting of neural activity to occur throughout the entire network, rather than as a separate appendage module.

Our analysis pipeline was validated by its overall agreement with commonly used alignment metrics (Kriegeskorte et al., 2008; Yamins et al., 2014; Raghu et al., 2017; Ostrow et al., 2023) in predicting the closest alignment at the recurrent stage. However, the reliability of such metrics has been questioned in the recent literature (Schaeffer et al., 2025; Anonymous, 2025; Bowers et al., 2023; Lampinen et al., 2025; Dujmovic et al., 2024; Serre, 2019b). Beyond this high-level alignment, our analysis also exposed some limitations of these alignment approaches, such as with the DSA metric (Ostrow et al., 2023). This highlights the advantage of our manifold-based framework over simple metric in-

Table 2: **Mean representational alignment metrics.** Mean taken over Representational Similarity Analysis (RSA), Canonical Correlation Analysis (CCA), Linear Predictivity (LP) and Dynamic Similarity Analysis (DSA) scores. Details in Appendix A.12); individual metric values in Table 6.

| Region | Enc L1 | Enc L2 | Enc L4 | Enc L5 | Enc L7 | Enc L8 | Rec | Readout | Output |
|--------|--------|--------|--------|--------|--------|--------|------|---------|--------|
| Retina | 0.26 | 0.26 | 0.30 | 0.33 | 0.28 | 0.28 | **0.40** | 0.34 | 0.34 |
| V1 | 0.29 | 0.21 | 0.32 | 0.30 | 0.30 | 0.32 | **0.53** | 0.38 | 0.48 |

spection: it provides a deeper understanding of the model's internal computations and representations. Tubularity was developed as a descriptive, data-driven characterization of population-level temporal organization. Rather than constituting an optimality principle for model design, it highlighted a salient structural property empirically present in biological recordings that was absent in early FNN layers.

**Future architecture improvements:** Our findings suggest several actionable insights for bringing foundation models, such as the FNN, into closer alignment with biological systems. (1) Coupling feature extraction with temporal dynamics: In biological systems, feature extraction and the development of temporal response dynamics occur simultaneously. Enforcing temporal dynamics in the early layers could enable more adequate modeling of the rich retinal dynamics. The FNN uses two temporally aware mechanisms in the recurrent module: attention and recurrence. We argue that recurrence, rather than attention, is the critical mechanism, as the FNN without attention yielded equal or better performance (Wang et al., 2025). Although our analysis was limited to the published attention-based version, we propose introducing early-stage recurrence that mimics amacrine cell connectivity in the retina (Marc et al., 2014). (2) Addressing padding-related artifacts: While padding is not an issue in biological systems, the resulting intensity artifacts can distort model representations. These artifacts are well known in convolutional architectures (Alsallakh et al., 2020), and could be addressed through alternative padding strategies, or tailored regularization, thereby freeing model capacity rather than requiring the readout to "unlearn" non-biological features. (3) Revising the readout stage: The current Gaussian readout layer (Lurz et al., 2021) combines a large number of feature maps through a single linear combination step, producing unrealistically distinct feature representations. Enforcing mixed features while reducing their number to better reflect biological cell type diversity (Bae et al., 2025) could push the representation towards smoother and more biologically realistic manifolds.

**Limitations:** Our analysis used a single foundation model, due to the limited availability of other video-based foundation models of neural activity over time. Moreover, we worked with a restricted set of stimuli (see Methods) to ensure comparability with biological data. However, there is evidence that these stimuli exercise much of the mouse visual cortex Dyballa et al. (2018), so they provide at least a necessary component for out-of-sample examination. Moreover, we show that these stimuli elicit activity patterns in the FNN similar to those evoked by the natural movies on which they were trained (Appendix Fig. 9), supporting their empirical validity. Finally, the tubularity metrics introduced here represent a novel approach for quantifying the geometry of neural trajectories. As no established methodological standards currently exist, further investigation of these metrics would be valuable. Specifically, systematic investigations of on both biological data and synthetic datasets would help assess robustness and for obtaining clear baselines. Additionally, incorporating curvature information could extend the metrics to capture additional characteristics of neural trajectories.

## 5    CONCLUSION

We found a rich diversity of encoding and decoding topologies in the FNN, highlighting its capability to fit complex neural data. Distinct representation patterns emerged across modules, reflecting its architecture. First, the *recurrent* module appears to learn generalizable representations of temporal stimuli, promoting uniformity and alignment, as in general self-supervised foundation models (Wang and Isola, 2022). Second, the *readout* module accounts for rich biological variability, but does so through a large number of self-similar feature maps, differing from the heterogeneous organization known in V1. Finally, the output layer achieves a continuous representation by linearly combining the readout features, ultimately enabling the network to associate spike trains with input movies *a posteriori*.

Using our novel tubularity metrics, we found that biological data exhibited strong stimulus-dependent structure in both retina and V1, whereas the FNN encoder trajectories lacked such tubularity. Only from the recurrent module onward did the FNN begin to form bundles of activity, reaching higher–though still sub-biological–levels of representational cohesion. This emphasizes the role of recurrence in generating biologically plausible temporal representations, suggesting that models may benefit from placing recurrence after a more light-weight, local encoder (e.g., emulating the amacrine connectivity in the retina (Marc et al., 2014)) and that constrain feature dimensionality to reflect biological cell-type diversity (Bae et al., 2025). While biological fidelity is not a prerequisite for achieving high predictive accuracy, digital-twin use cases require enough internal alignment to support mechanistic and interventional inference. Such designs could help bridge the gap between computational performance and biological plausibility, moving toward truly brain-aligned foundation models.

# 6 ETHICS STATEMENT

There are no ethical concerns for this paper.

# 7 REPRODUCIBILITY STATEMENT

We provide an overview of our methods in the main text (Section 2) and include further details for reproducing our results in the Appendix A. Upon acceptance, we will make the code for all experiments and figures available on GitHub.

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

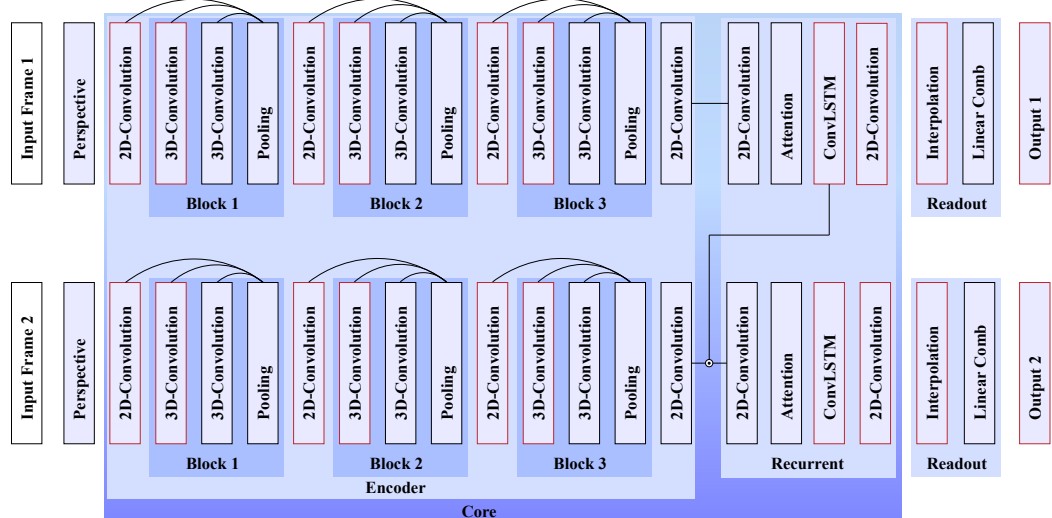

Figure 6: **FNN architecture.** Layers used for sampling are highlighted. Modulation module omitted as it has no effect for our analysis. The FNN used GeLU activations in the convulutional layers, and Tanh activations in the Recurrent module.

# Appendix

## A    METHODS

### A.1    ONLINE MATERIAL

Our work made use of publicly available open-source resources. Specifically, we employed the pretrained FNN model provided by Wang et al. (2025), available at `https://github.com/cajal/fnn/tree/main`. For the analysis of this model, we used the stimulus generation tools and neural encoding manifold construction pipeline introduced by Dyballa et al. (2024a), accessible at `https://github.com/dyballa/NeuralEncodingManifolds`.

### A.2    FNN

The FNN consists of five modules: perspective, modulation, *encoder*, *recurrent*, and *readout* (Fig. 6). The perspective and modulation modules model the mouse's state and transform the inputs to approximate the actual visual information received. Thus, only the *encoder*, *recurrent*, and *readout* modules perform the core computation and are the focus of this work.

The *encoder* module is a 10-layer DenseNet-style convolutional encoder. Notably, it includes 3D convolutions, which in principle enable the encoder to capture temporal patterns, for up to 12 time steps into the past for later encoder layers. The *recurrent* module is optionally preceded by an attention layer and consists of a convolutional LSTM, followed by a single convolutional layer that produces its output. This feedforward–recurrent combination constitutes the core of the FNN, which is trained on all data. Finally, the *readout* module is mouse-specific: it performs an interpolation on the recurrent output followed by a linear transformation to produce the FNN output. We used the FNN readout from session 8, scan 5 as this was the exemplary scan used in the authors' tutorial. We validated findings on several other sessions and scans.

### A.3    INPUT VIDEOS

We used the visual stimuli from Dyballa et al. (2024a), consisting of drifting square-wave gratings and optical flows moving in eight directions. The flow stimuli include oriented (lines) and non-oriented (dots) stimuli with spatial frequencies between 0.04 and 0.5 $\frac{\text{cycles}}{\text{deg}}$. This yields 88 unique input

sequences with stochastic initial positions and velocities. The stimuli were scaled and cropped to fit the required FNN input shape of $144{\times}256$ pixels. This resulted in an image sequence: $\{\mathbf{x}_0, \ldots, \mathbf{x}_T\}$, where each $\mathbf{x}_i \in \mathbb{R}^{H \times W}$. Stimuli were generated using the tools available at `https://github.com/dyballa/NeuralEncodingManifolds`.

The FNN (Wang et al., 2025) processes 2.33-second sequences of 70 frames each, corresponding to 30 frames per second. Since in Dyballa et al. (2024a) the trials were 1.25 s long, we adapted the stimuli to contain 37 frames to maintain consistency with the FNN framework. This adaptation was performed using the hyperparameters of the stimulus generation pipeline, allowing comparable stimuli dynamically created for different lengths and number of frames.

We acknowledge a difference in the experimental setups regarding the visual field: Wang et al. (2025) used a screen distance of 15 cm, whereas the stimuli from Dyballa et al. (2024a) were originally designed for a 25 cm viewing distance. This discrepancy potentially affects the visual field transformations performed by the model's perspective module, as the visual angle subtended by the stimuli differs between the two configurations. We applied a global scaling factor of 0.7 to all stimuli to address this. This adjustment was empirically found to optimize stimulus discriminability across network layers, effectively bridging the geometric gap between the training and analysis domains.

## A.4 DATA SAMPLING

Neural responses were computed using PyTorch and extracted by sampling activations from 2000 units across selected FNN layers. Within each layer, 40 feature maps were sampled. Then, 50 neurons were sampled from each feature map. Feature map sampling probabilities were calculated from the mean maximum response across all neurons within each map, while neuron sampling probabilities within each selected feature map were based on individual neuron maximum responses, biasing the sampling to include active neurons. This sampling procedure was chosen to ensure comparability to the biological results from Dyballa et al. (2024a). This sampling procedure was tested and validated in Dyballa et al. (2024a); we performed further tests with random sampling to validate this bias does not filter out relevant structures. One exemplary sampling result, showing qualitative stability of results across sampling strategies and sizes can be found in Fig 18. Increasing the sampling rate beyond 2000 units did not significantly alter manifold topology but hindered cluster separation in diffusion map analysis. The resulting tensor data had dimensions $(N \times S \times O \times T)$ with $N = 2000$ neurons, $S = 11$ stimulus types, $O = 8$ orientations and $T = 37$ time steps. For manifold construction, the optimal spatial frequency was selected (resulting in $S = 6$ stimuli) whereas for classification performance all spatial frequencies were kept. We report results from a single random seed per layer, as preliminary analysis showed consistent manifold structure across different random activity samples. These neural activation tensors served as input for subsequent classification and manifold analysis. This sampling procedure was developed by Dyballa et al. (2024a) and tested against other sampling methods there. We also experimented with the sampling procedure, finding that random sampling and increased sampling rate did not introduce qualitative changes to the manifolds.

## A.5 STIMULUS ADEQUACY

For every FNN layer investigated in this paper, we extracted the activation to the stimulus ensemble consisting of gratings and flows (see Section A.3) as well as to a 100-second-long natural input video from the MICrONS functional dataset (Bae et al., 2025), downloaded from `s3://bossdb-open-data/iarpa_microns/minnie/functional_data/stimulus_movies/`. Both stimulus sets produced similar activation magnitudes across the entire network (see Fig. 9), which shows the adequacy of the stimulus ensemble used for testing the FNN.

For orientation and direction selectivity, we followed Wang et al. (2025)'s procedure: We input directional pink noise (16 directions, 37 frames) to the model and record the output activations. Additionally, we recorded the outputs for our flow stimuli. For both datasets, we computed the Orientation Selectivity Index (OSI) and Direction Selectivity Index (DSI) and compared their distributions.

$$OSI = \frac{|\sum_\theta \overline{r}_\theta e^{i2\theta}|}{\sum_\theta \overline{r}_\theta} \qquad (1)$$

$$DSI = \frac{|\sum_\theta \overline{r}_\theta e^{i\theta}|}{\sum_\theta \overline{r}_\theta} \qquad (2)$$

Here, $\overline{r}_\theta$ is the mean response for angle $\theta$. We found comparable OSI and DSI distributions (see Fig. 10).

## A.6 CLASSIFICATION ACCURACY

Classification performance of the stimulus set, measured using activations, serves as a proxy for representational richness. Logistic regression is employed to assess linear separability, while k-Nearest Neighbor (k-NN) classification is used to evaluate local geometric structure for comparison with logistic regression.

Stimulus classification accuracy based on individual-layer activities was determined by training multinomial logistic regression classifiers (solver: L-BFGS) with 5-fold cross-validation (CV). Only sampled neurons were used to classify the 11 stimuli. For each layer and each time point t, two feature sets were constructed: (i) the mean activity over frames 0 to t (increasing window) and (ii) the mean activity over frames t to end (decreasing window). For comparison, K-nearest neighbor classifiers (K=3) were also evaluated using leave-one-out CV. The value K=3 was selected as the optimal neighborhood size. Leave-one-out CV was used for k-NN due to its suitability for small datasets, while 5-fold CV was chosen for logistic regression to reduce computational requirements. Results are summarized in Table 1 and Fig. **??**.

## A.7 CONSTRUCTION OF DECODING MANIFOLDS

For building the *decoding manifolds*, we applied PCA (scikit-learn) to the averaged activity data. In total, the *decoding manifolds* contain 48 points, consisting of 6 stimuli and 8 movement directions each. The 6 stimuli were obtained from a majority vote of all neurons on the optimal spatial frequency eliciting higher responses. The decoding manifolds use different colors for each stimulus, as introduced in Fig. 1. Different spatial frequencies of the same stimulus are summarized with the same color. To construct *decoding trajectories*, we treated each time step as a separate data point rather than averaging across time before applying PCA. In both cases, we reduced the dimensionality to three components for visualization after verifying that further dimensions did not encode qualitatively new information. We constructed biological *decoding trajectories* using experimental data from Dyballa et al. (2024a), available at `https://github.com/dyballa/NeuralEncodingManifolds`. For the biological decoding trajectories, we did not use the additional zero-activity time step since a baseline activity level was already provided by the inter-stimulus intervals in the experiments.

## A.8 TUBULARITY

Before calculating tubularity metrics, we standard-scale the data and apply PCA to obtain a 10-dimensional embedding, thereby speeding up the computation. While visualizations use only the first 2–3 dimensions, all metrics are calculated in the 10-dimensional space. To ensure comparability, we resampled all trajectories to length 100. For statistical analysis, we generated 100 bootstrapped samples, and using ground-truth clusters, performed Bonferroni-corrected Mann-Whitney U tests on our hypotheses.

We formalize how "tight" a group of curves is around the centerline: We reparameterize each curve by normalized arc length $u \in [0, 1]$ and resample to $\{u_k\}_{k=1}^M$. Let $x_i(u_k) \in \mathbb{R}^D$ denote the samples and $\tau_i(u_k)$ their unit tangents. We define the *mean curve* as the pointwise average:

$$c(u_k) \ = \ \frac{1}{m}\sum_{i=1}^m x_i(u_k), \qquad r_i(u_k) \ = \ \|x_i(u_k) - c(u_k)\|.$$

The tightness score is calculated by averaging quantile tube radii across bins $\{I_b\}_{b=1}^{B}$ that partition $[0, 1]$, using a high quantile $q \in [0.8, 0.95]$ to ensure robustness to noise. We normalize each tube's tightness score by tube length.

$$S_{\text{tight}} \;=\; \frac{1}{B} \sum_{b=1}^{B} \text{quantile}_q \{\, r_i(u) : u \in I_b \text{ over all curves} \,\}.$$

The second quantity assessed is the uniformity of the tubes relative to one another. That is, the degree to which crossings occur in our defined bundle of curves. Tubes are considered disorganized when distinct curves pass near each other with *transverse* directions. Let $d_{ij}(u, v) = \|x_i(u) - x_j(v)\|$ and $\phi_{ij}(u, v) = 1 - \langle \tau_i(u), \tau_j(v) \rangle^2 \in [0, 1]$ (large for near-orthogonal tangents). Using a Gaussian kernel $K_\varepsilon(\rho) = \exp(-\rho^2/(2\varepsilon^2))$, we softly count encounters:

$$\mathcal{X}_\varepsilon = \frac{2}{m(m-1)} \sum_{i<j} \int_0^1 \int_0^1 K_\varepsilon\big(d_{ij}(u,v)\big)\, \phi_{ij}(u,v)\, du\, dv.$$

$S_{\text{tight}}$ and $S_{\text{cross}}$ only depend on distance, unit-tangent inner product, and arc-length. Therefore, they are invariant to translations, rotations, and re-timing. We emphasize that, for both scores, smaller values indicate more tubular curve bundles, while larger values indicate fewer tubular curve bundles.

A.9    Construction of neural encoding manifolds

At a high level, the motivation for constructing *neural encoding manifolds* is to find a space in which one can examine the global topology of neuronal populations based on their stimulus selectivities and temporal response patterns (Dyballa et al., 2024a). The neural encoding manifold is constructed in a three-step procedure. First, a 3-tensor is built with the temporal responses from each neuron for each stimulus, and decomposed using Nonnegative Tensor Factorization (details below); each component is comprised of neural, stimulus, and temporal response factors. The neural factors then serve as position coordinates, embedding the neurons into a stimulus-response framework called the neural encoding space. Second, we construct a data graph in this neural encoding space using the IAN algorithm (Dyballa and Zucker, 2023). Third, applying diffusion maps (Coifman et al., 2005; Coifman and Lafon, 2006) to the data graph yields the manifold.

The methodological choices in our manifold construction procedure are made in accordance with Dyballa et al. (2024a), where extensive parameter analysis for biological neural data was conducted. Since *neural encoding manifolds* computed with these specific parameters represent the only available comparison for biological data from the visual system, we maintained their parameter settings to ensure direct comparability between artificial and biological neural representations. We further conducted analysis for FNN-specific parameters, such as the sampling procedure, by adapting their code to fit the FNN requirements.

A.9.1    Preprocessing

The input tensor of neuronal activity (see above) was preprocessed in several steps (using NumPy and SciPy). First, the individual responses were smoothed along the time dimension using a one-dimensional Gaussian kernel with $\sigma = 3$. Next, we grouped the stimuli into *medium* versus *high* spatial frequencies and selected the one exhibiting higher response magnitudes. The temporal responses for the 8 directions of motion were then concatenated together into a single vector. Finally, we normalized each response and rescaled it by the relative activations of the neuron. The resulting tensor $\mathbf{T}$ had shape $((N = 2000) \times (S = 6) \times (O * T = 296))$.

A.9.2    Nonnegative Tensor Factorization

Next, Nonnegative Tensor Factorization (see (Williams et al., 2018) for an overview and applications to neuroscience) was applied to our tensor $\mathbf{T}$. It was decomposed into typically 10–15 rank-1 tensors which are obtained from the outer product of three vectors each. We selected the number of components separately for each data sample based on changes in explained variance and noise,

following the procedure in Dyballa et al. (2024a). The factors in each component are scaled to unit length, and their magnitudes absorbed by a scalar $\lambda_r$:

$$\tilde{\mathbf{T}} = \sum_{r=1}^{R} \lambda_r \mathbf{v}_r^{(1)} \circ \mathbf{v}_r^{(2)} \circ \mathbf{v}_r^{(3)} = [\lambda; \mathbf{X}^{(1)}; \mathbf{X}^{(2)}; \mathbf{X}^{(3)}] \tag{3}$$

For the second equality, the factor matrices $\mathbf{X}^{(k)}$ are constructed using the factor vectors $\mathbf{v}_r^{(k)}$ as columns, and the vector $\lambda$ contains all individual $\lambda_r$s.

Decomposing the tensor $\mathbf{T}$ into these components is an optimization problem with the following objective function and non-negativity constraints:

$$\min_{\mathbf{X}^{(1)}, \mathbf{X}^{(2)}, \mathbf{X}^{(3)}} \frac{1}{2} ||\mathbf{T} - \tilde{\mathbf{T}}||^2 \tag{4}$$

$$\text{such that} \quad \mathbf{X}^{(k)} \geq 0, \forall k \tag{5}$$

The resulting decomposition is interpretable: the third group of vectors, $\mathbf{v}_r^{(3)}$, describes different temporal response patterns; $\mathbf{v}_r^{(2)}$ contain information about which stimuli exhibit these response patterns; and $\mathbf{v}_r^{(1)}$ are the neuronal factors determining which neurons exhibit the response patterns characterized by $\mathbf{v}_r^{(2)}$ and $\mathbf{v}_r^{(3)}$. During decomposition, circular permutations were applied to detect patterns irrespective of the preferred orientations of specific neurons (again, this is necessary to ensure compatibility with the biological results from (Dyballa et al., 2024a)).

Using the OPT method from Tensor Toolbox (Bader et al., 2023)), we ran the decomposition 50 times (different initializations) for each number of components and dataset to ensure robust decomposition results and the choice of the number of factors, $R$. The manifolds were robust to small changes in $R$, therefore the heuristic for choosing $R$ based on the explained variance of the decomposition outlined in Dyballa et al. (2024a) proved sufficient. For building the manifolds, we used the result with smallest reconstruction error among the 50 initializations.

### A.9.3 NEURAL ENCODING SPACE

Following Dyballa et al. (2024a), we now reformulate the above decomposition to construct the neural encoding space. By defining the diagonal matrix $\mathbf{\Lambda}$ with $\mathbf{\Lambda}_{rr} = \lambda_r$, we obtain:

$$\tilde{\mathbf{T}} = \mathbf{X}^{(1)} \mathbf{\Lambda} (\mathbf{X}^{(2)} \circ \mathbf{X}^{(3)}) \tag{6}$$

Since the first matrix, $\mathbf{X}^{(1)}$, represents the neuronal factors, we denote it by $\mathcal{N}$. Now, define a matrix $\mathbf{B}$ with columns $\mathbf{b}_{:,r}$:

$$\mathbf{b}_{:,r} = vec(\mathbf{v}_r^{(2)} \circ \mathbf{v}_r^{(3)}) \tag{7}$$

Finally, we obtain a matrix representation of $\mathbf{T}$ with respect to neuronal factors as $\mathbf{X}_{\mathcal{N}}$:

$$\mathbf{X}_{\mathcal{N}} = \mathbf{B} \mathbf{\Lambda} \mathcal{N}^T \tag{8}$$

This reformulation constructs the neural encoding space. The unit-norm basis vectors of this space are given by the columns of $\mathbf{B}$. We define the neural matrix containing the positions of all neurons in this space as $\mathcal{N}_\lambda = \mathcal{N} \mathbf{\Lambda}$. The distances between any two neurons in this space reflect their similarity in stimulus-selective temporal response patterns. Intuitively, neurons with similar selectivity profiles and temporal dynamics should be positioned close together, while neurons with dissimilar response characteristics should be farther apart.

### A.9.4 Iterated adaptive neighborhoods (IAN)

Within this neural encoding space, we construct a weighted graph of the data by inferring a similarity kernel. This is achieved using the Iterated Adaptive Neighborhoods (IAN) algorithm (Dyballa and Zucker, 2023), which infers an adaptive local kernel without the need for pre-specifying a fixed neighborhood size.

IAN first constructs the unweighted Gabriel graph for the data points. In addition, a weighted graph is constructed using a multiscale Gaussian kernel based on the discrete neighborhood graphs. Subsequently, the graph is iteratively pruned by ensuring consistency between the discrete and continuous neighborhoods. The resulting weighted graph is represented by the adjacency (kernel) matrix $\mathbf{K}$. This matrix contains similarities computed using locally tuned Gaussian kernels.

### A.9.5 Diffusion maps

Diffusion Maps (Coifman et al., 2005; Coifman and Lafon, 2006) are a dimensionality reduction technique that retain distances and preserve the intrinsic geometry of the manifold. The diffusion process is based on graph Laplacian normalization from spectral graph theory.

In detail, we use the weighted graph obtained from IAN as the weighted adjacency matrix $\mathbf{K}$. The first step is to normalize and symmetrize it to produce $\mathbf{M}_s$:

$$\mathbf{d}_i = \sqrt{\sum_j \mathbf{K}_{ij} + \epsilon} \tag{9}$$

$$\mathbf{M}_s = \frac{\mathbf{K}}{\mathbf{dd}^T} \tag{10}$$

This normalization ensures that nodes of high degree do not dominate the analysis. We then calculate the spectral decomposition of $\mathbf{M}_s$ with eigenvalues $\lambda_0 = 1 \geq \lambda_1 \geq \lambda_2...$ and eigenvectors $\boldsymbol{\psi}_i$ for $t = 1$ diffusion steps using $L = 20$ eigenvalues:

$$\mathbf{M}_{s,ij}^t = \sum_{l=0}^{L} \lambda_l^{2t} \boldsymbol{\psi}_l(i) \boldsymbol{\psi}_l(j) \tag{11}$$

Finally, from the spectral decomposition, we obtain the diffusion map with diffusion coordinates:

$$\Psi_t(i) = \begin{pmatrix} \lambda_0^t \boldsymbol{\psi}_0(i) \\ \lambda_1^t \boldsymbol{\psi}_1(i) \\ \vdots \\ \lambda_{L-1}^t \boldsymbol{\psi}_{L-1}(i) \end{pmatrix} \tag{12}$$

Plotting the data using these diffusion coordinates yields the *neural encoding manifold*.

### A.9.6 Encoding manifold visualization

For visualization purposes, we optionally applied metric multidimensional scaling (MDS) to the diffusion map coordinates. This was done by computing pairwise squared Euclidean distances using the first diffusion coordinates, constructing the corresponding Gram matrix $\mathbf{G} = -0.5 * \mathbf{D}^2$, and applying kernel PCA to obtain a lower-dimensional embedding. This preserves the distance relationships from the diffusion map while combining multiple diffusion coordinates, enabling a clearer visualization of the manifold structure.

Based on the manifold topology, we selected groups of neurons to investigate via their PeriStimulus Time Histograms (PSTH). We averaged their activity across trials and constructed the PSTHs as a 2-D heatmap, where each row contains the temporal activity in response to a particular direction of motion (as displayed in Fig. 1). Additionally, we calculated the average response intensity over time for these groups and reported the s.e.m. using the shaded regions (see insets in Fig. 2A,D).

### A.10 VISUALIZATIONS

Interactive three-dimensional plots of the manifolds were computed using Plotly. Other plots were created with Matplotlib and TUEplots.

### A.11 MINIMODELS

For our additional analysis in Fig. 17, we used the convolutional model introduced in Du et al. (2025). We downloaded model checkpoints from `https://github.com/MouseLand/minimodel/tree/main`. We left the manifold pipeline unchanged for this experiment and sampled activations from layer 2.

### A.12 ALIGNMENT METRICS

For comparability, the biological data was downsampled to 37 time steps for all alignment metric calculations. Except for DSA, all metrics were calculated on the individual time steps and the averaged.

#### A.12.1 REPRESENTATIONAL SIMILARITY ANALYSIS (RSA)

RSA (Kriegeskorte et al., 2008) is computed by obtaining the Representational Dissimilarity Matrices (RDMs) for every time step individually via $RDM = 1 - Pearson Correlation$. Then, based on the upper triangular values (excluding diagonals), the RSA scores are obtained from the Spearman's r (using scipy stats) between biological and artificial data.

#### A.12.2 CANONICAL CORRELATION ANALYSIS (CCA)

For CCA (Raghu et al., 2017), the data was first dimensionality reduced using PCA (3 components). Then, using sklearn's CCA function, the first 3 canonical vectors were obtained and their correlations between brain and model were averaged, yielding CCA.

#### A.12.3 LINEAR PREDICTIVITY (LP)

Linearly predicting individual biological neurons from FNN data using Ridge Regression did not yield adequate scores due to the high amount of noise. Therefore, we again used PCA to obtain 3 components for brain data and 20 components for artificial data. We then fit Ridge Regression ($\alpha = 1$ to predict individual components of biological data using 40 random stimuli, and measured the average performance on the 8 heldout stimuli via $R^2$ (Yamins et al., 2014).

#### A.12.4 DYNAMICAL SIMILARITY ANALYSIS (DSA)

For DSA (Ostrow et al., 2023), we again simplified data using PCA (10 components) and computed DSA scores using the DSA function from the authors. We reported inverted DSA scores ($1 - DSA$) to compare with other metrics, and Z-scores compared to a null distribution of 50 samples where the time steps of FNN data were randomly shuffled before comparing to biology.

#### A.12.5 CRITIQUE

The validity of these methods for comparing brains and machines has been questioned (Serre, 2019b). Schaeffer et al. (2025) argue that lower LP scores may correspond to less brain-like models, as, instead of selecting biological models, LP overfits biases in linear regression. They claim that the same holds for overfitting to other representational similarity metrics. It is unclear what brain alignment means and what the different alignment metrics truly measure (Anonymous, 2025). Metric variability can fall within individual subject variability, making clear conclusions difficult (Anonymous, 2025). Bowers et al. (2023) question the assumption of biological visual systems being optimized to classify objects. Also, they claim that differences in the features used in DNNs compared to those in the brain can lead to high similarity. Moreover, simple features can be overrepresented compared to complex features, biasing the similarity scores (Lampinen et al., 2025). Finally, Dujmovic et al. (2024) find that metrics like RSA are not robust with respect to input perturbations.

## A.13 SOFTWARE

All software (Table 3) is used in accordance with its respective license.

Table 3: Software packages used in this work.

| Package | Version | License |
|---|---|---|
| MATLAB Tensor Toolbox (Bader et al., 2023) | 3.6 | BSD-2 |
| IAN (Dyballa and Zucker, 2023) | 1.1.2 | BSD-3 |
| NeuralEncodingManifolds (Dyballa et al., 2024a) | N/A | BSD-2 |
| NumPy (Harris et al., 2020) | 1.25.0 | BSD-3 |
| SciPy (Virtanen et al., 2020) | 1.15.3 | BSD-3 |
| scikit-learn (Pedregosa et al., 2011) | 1.7.1 | BSD-3 |
| PyTorch (Paszke et al., 2019) | 2.6.0 | MIT |
| Matplotlib (Hunter, 2007) | 3.10.1 | PSF-based (BSD-compatible) |
| Plotly (Inc., 2015) | 6.0.0 | MIT |
| TUEplots (Krämer et al., 2024) | 0.2.0 | MIT |

## A.14 COMPUTE

The experiments were conducted on an HPC cluster. FNN sampling uses randomly selected GPUs (RTX 2080 Ti, or better). All other experiments were performed on CPU. All experiments required less than 30 GB memory. In total, 10 tensor decomposition experiments were run on CPU, each taking 2 days on a single CPU. Preliminary results not included in the paper required another 50 tensor decomposition experiments.

## A.15 LANGUAGE MODEL USAGE

At the level of individual words or partial sentences, language models were used to fix language errors. Minor code sections were produced by language models and used only after careful inspection.

## B DATA AND CODE AVAILABILITY

Upon acceptance, we will publish a GitHub repository with the full code necessary to reproduce all experiments and figures in this paper. We will also provide rotating video animations of three-dimensional visualizations to aid interpretation.

## C SUPPLEMENTAL FIGURES

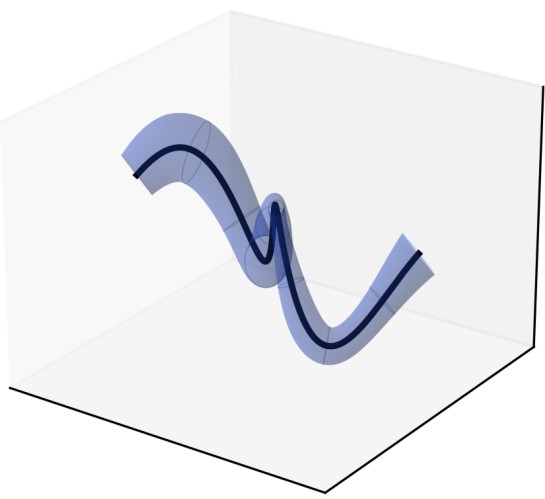

Figure 7: A tubular neighborhood around a centerline $c(u)$ with radius profile $R(u)$.

Table 4: **Tubularity metrics for biological and FNN data**. Low tightness and crossings values indicate high tubularity. Aligning with the visualization in Fig. **??**, the biological trajectories show highly tubular organizations compared to FNN. Method details in Appendix **??**.

| Layer | Ground Truth Labels | | HDBSCAN Labels | | |
|---|---|---|---|---|---|
| | $S_{\text{tight}}$ | $S_{\text{cross}}$ | $S_{\text{tight}}$ | $S_{\text{cross}}$ | **Clusters** |
| Retina | 0.0688 | $1.29 \times 10^{-05}$ | 0.1017 | $1.06 \times 10^{-05}$ | 4 |
| V1 | 0.1357 | $4.06 \times 10^{-05}$ | 0.1859 | $3.50 \times 10^{-05}$ | 4 |
| Enc1 | 0.2018 | $2.87 \times 10^{-04}$ | 0.7680 | $1.66 \times 10^{-04}$ | 1 |
| Enc13 | 1.9885 | $1.77 \times 10^{-06}$ | 4.3461 | $1.09 \times 10^{-06}$ | 3 |
| Rec | 0.1228 | $2.65 \times 10^{-07}$ | 0.1697 | $1.53 \times 10^{-07}$ | 4 |
| RecOut | 0.1209 | $5.72 \times 10^{-07}$ | 0.1650 | $5.34 \times 10^{-07}$ | 4 |
| Readout | 0.3307 | $3.96 \times 10^{-06}$ | 0.4320 | $5.40 \times 10^{-06}$ | 4 |
| Output | 0.1483 | $3.53 \times 10^{-06}$ | 0.2784 | $1.12 \times 10^{-06}$ | 3 |

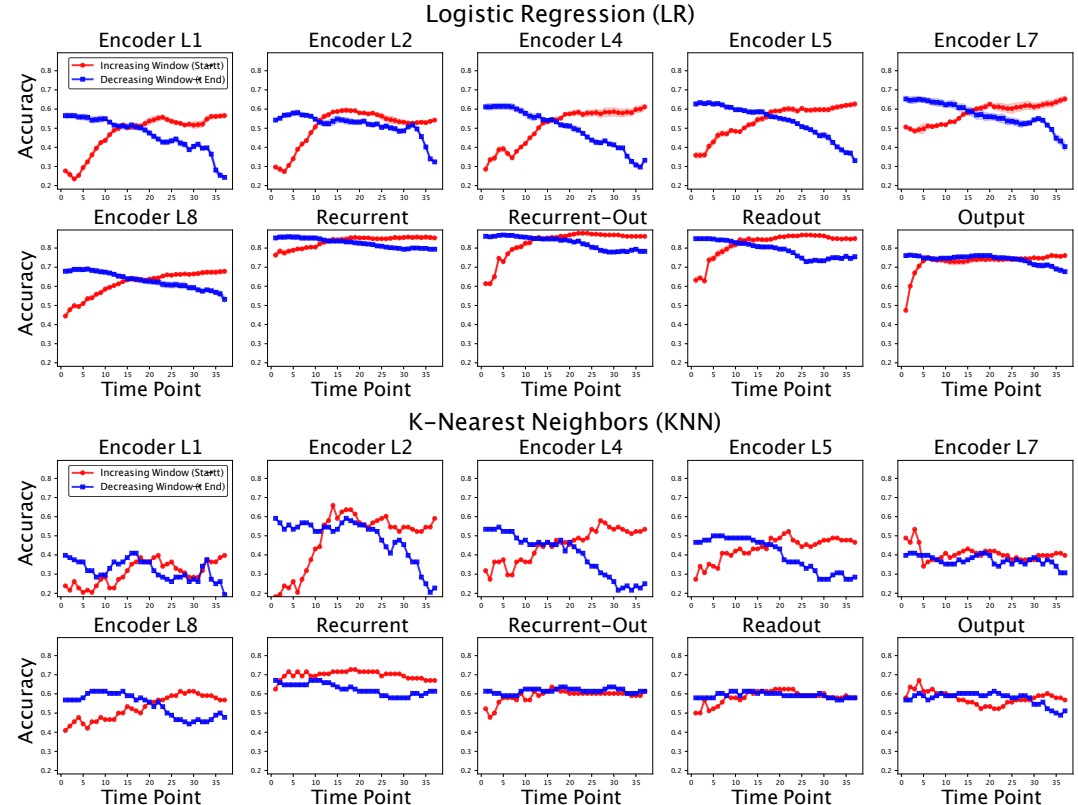

Figure 8: **Logistic regression (LR, top) and K-Nearest Neighbor (KNN, K=3, bottom) classifier accuracy** for each layer. We use increasing time windows (timesteps $0 \rightarrow t$, red) or decreasing time windows ($t \rightarrow 37$, blue) to calculate the accuracies. Shaded regions for LR show the s.e.m. The maxima across panels are summarized in Table 1.

Table 5: Representational Similarity Analysis (RSA), Canonical Correlation Analysis (CCA), Linear Predictivity (LP) and Dynamic Similarity Analysis (DSA) scores and DSA Z-scores to a time-shuffled baseline. High values indicate closer alignment for all metrics. (see Appendix A.12)

| Region | Metric | L1 | L2 | L4 | L5 | L7 | L8 | Rec | Readout | Output |
|--------|--------|------|------|------|------|------|------|------|---------|--------|
| Retina | RSA | -.04 | -.03 | -.03 | -.01 | -.01 | 0.03 | 0.03 | **0.05** | -.01 |
| Retina | CCA | 0.19 | 0.25 | 0.25 | 0.30 | 0.27 | 0.26 | **0.34** | 0.25 | 0.32 |
| Retina | LP | 0.05 | 0.06 | 0.17 | 0.29 | 0.26 | 0.28 | **0.43** | 0.24 | 0.26 |
| Retina | DSA | **0.84** | 0.77 | 0.83 | 0.73 | 0.58 | 0.56 | 0.80 | 0.81 | 0.80 |
| Retina | DSA-Z | **5.90** | 4.77 | 5.29 | 4.55 | 2.35 | 1.87 | 2.00 | 1.79 | 4.45 |
| V1 | RSA | -.11 | -.27 | -.16 | -.22 | 0.10 | 0.08 | **0.46** | 0.08 | 0.41 |
| V1 | CCA | 0.29 | 0.31 | 0.33 | **0.39** | 0.29 | 0.35 | **0.39** | 0.29 | 0.38 |
| V1 | LP | 0.05 | 0.04 | 0.18 | 0.29 | 0.23 | 0.27 | **0.40** | 0.22 | 0.24 |
| V1 | DSA | 0.91 | 0.76 | **0.93** | 0.76 | 0.59 | 0.57 | 0.88 | 0.92 | 0.90 |
| V1 | DSA-Z | **7.03** | 5.49 | 6.52 | 5.91 | 5.46 | 4.40 | 4.72 | 3.50 | 5.27 |

Table 6: Explained Variance (EV, in %) of decoding manifold PCA, and number of tensors (R) and Error Percentage (EP) of tensor factorization (in %)

| Metric | L1 | L2 | L4 | L5 | L7 | L8 | Rec | RecOut | Readout | Output |
|--------|------|------|------|------|------|------|------|--------|---------|--------|
| EV | 47.91 | 57.66 | 59.52 | 48.20 | 48.53 | 42.77 | 57.73 | 53.43 | 53.99 | 55.23 |
| R | 8 | 12 | 11 | 11 | 13 | 11 | 9 | 12 | 17 | 14 |
| EP | 37.08 | 25.50 | 23.02 | 23.35 | 22.34 | 23.51 | 15.68 | 16.45 | 13.81 | 9.26 |

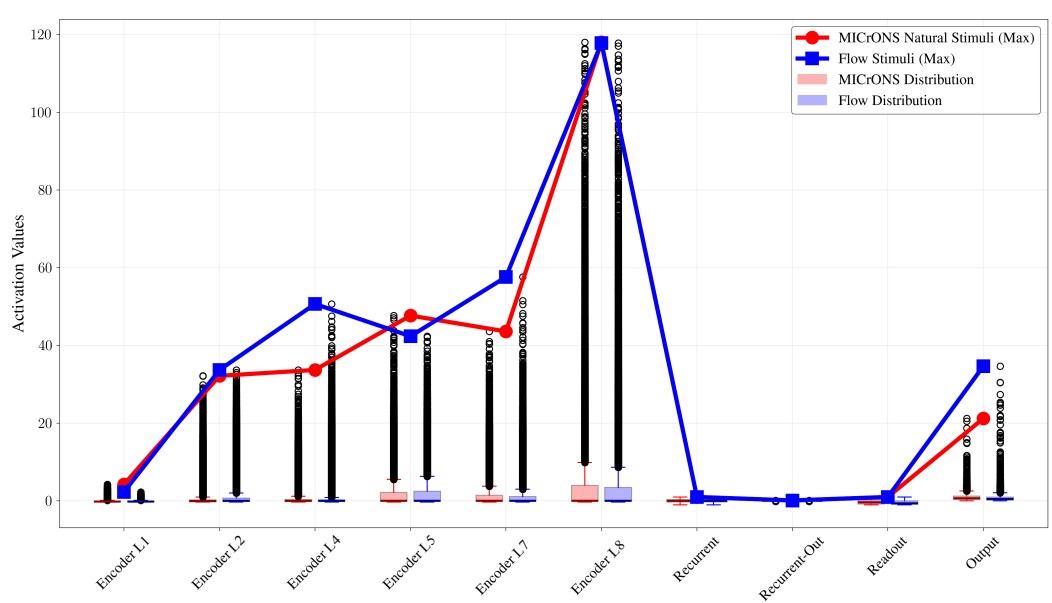

Figure 9: **Activation function output distributions and maxima** for natural MICrONS (Bae et al., 2025) input videos and the flow stimulus ensemble (Dyballa et al., 2024a). The comparable activity across network layers shows the adequacy of investigating the FNN with flow stimuli. The differences in magnitudes across layers are explained by the activations functions (GELU in the *encoder*, Tanh in the *recurrent* and *readout* modules).

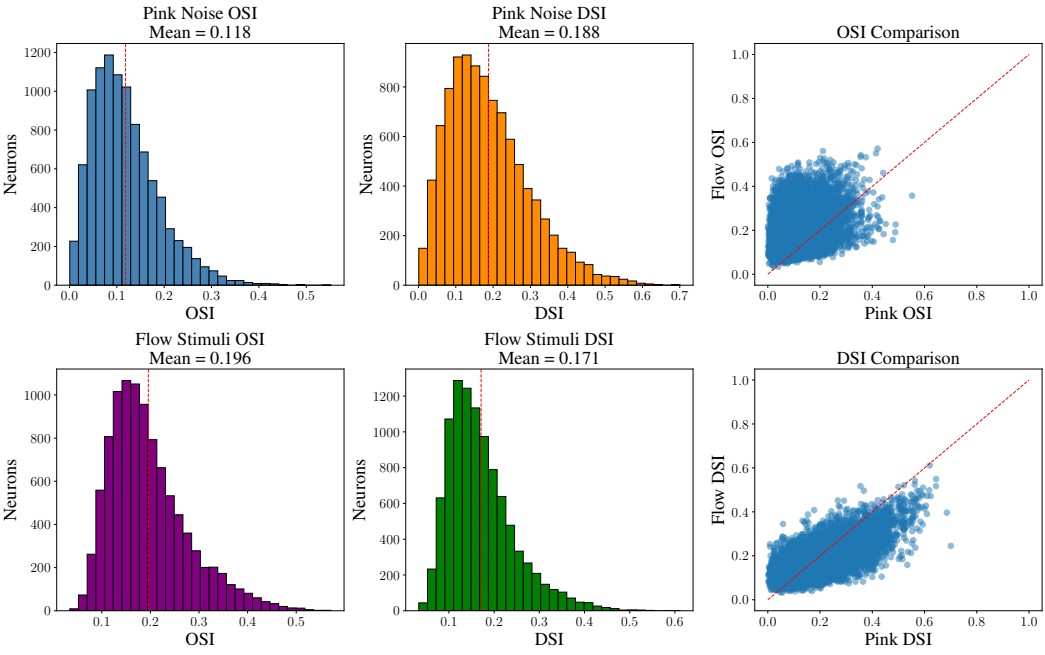

Figure 10: **OSI and DSI of FNN output** for pink noise (as used in Wang et al. (2025)) and for the stimulus ensemble from Dyballa et al. (2024a), meaned over the different stimuli.

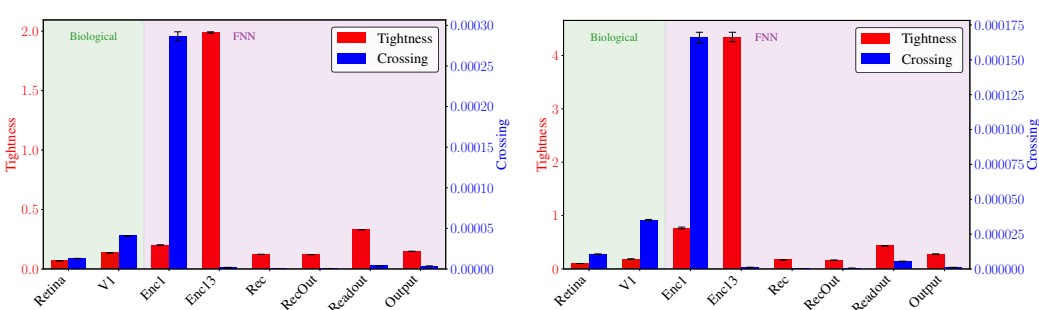

Figure 11: **Tubularity comparison between biological and FNN data.** Tightness, measuring how close trajectories within a bundle are to their centerline, and crossings, measuring the amount of transverse crossings in a bundle, are scores for biological and FNN data. Left: Using ground-truth stimulus class labels. Right: Using HDBSCAN (Campello et al., 2013) labels.

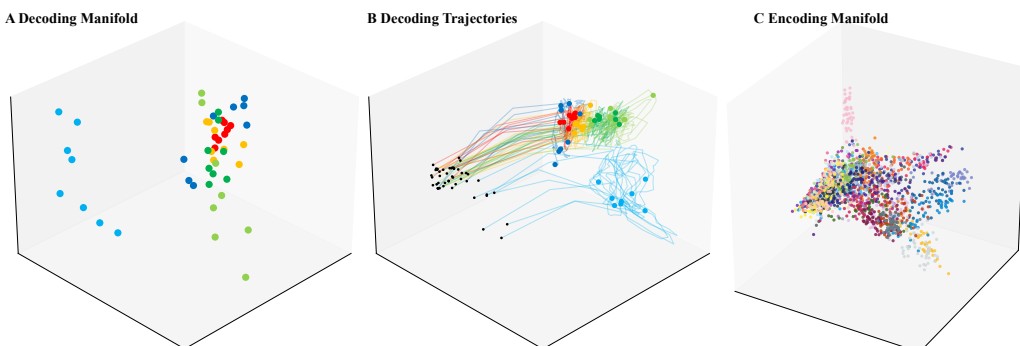

Figure 12: **Encoder L2.** Note the encoding manifold smoothness here results from the early layer only capturing simple features and the dominance of intensity discussed for L8. We therefore do not interpret this as a V1-like smooth encoding manifold.

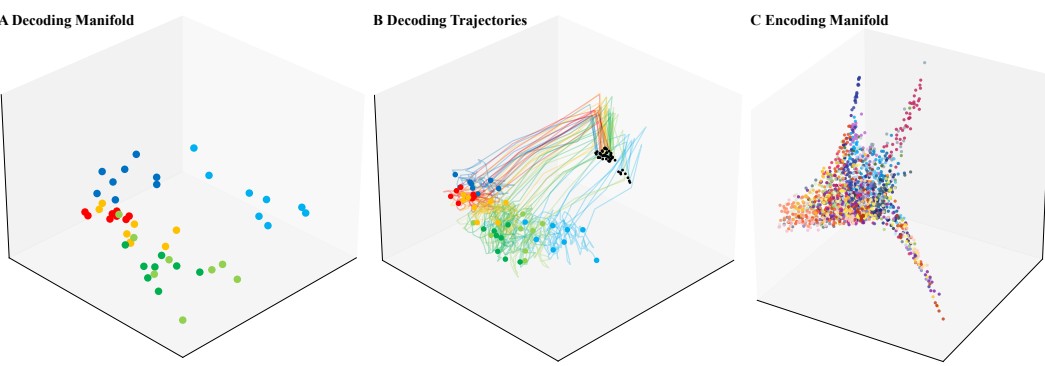

Figure 13: **Encoder L4.** Note the encoding manifold smoothness here results from the early layer only capturing simple features and the dominance of intensity discussed for L8. We therefore do not interpret this as a V1-like smooth encoding manifold.

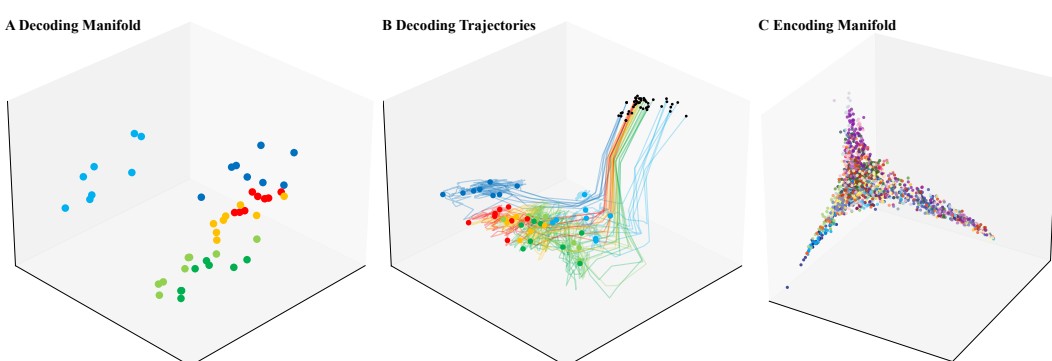

Figure 14: **Encoder L5.** Note the encoding manifold smoothness here results from the early layer only capturing simple features and the dominance of intensity discussed for L8. We therefore do not interpret this as a V1-like smooth encoding manifold.

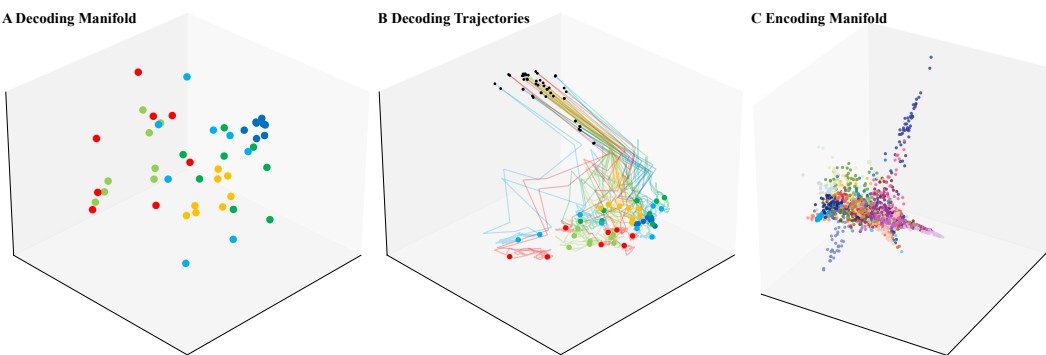

Figure 15: **Encoder L7**

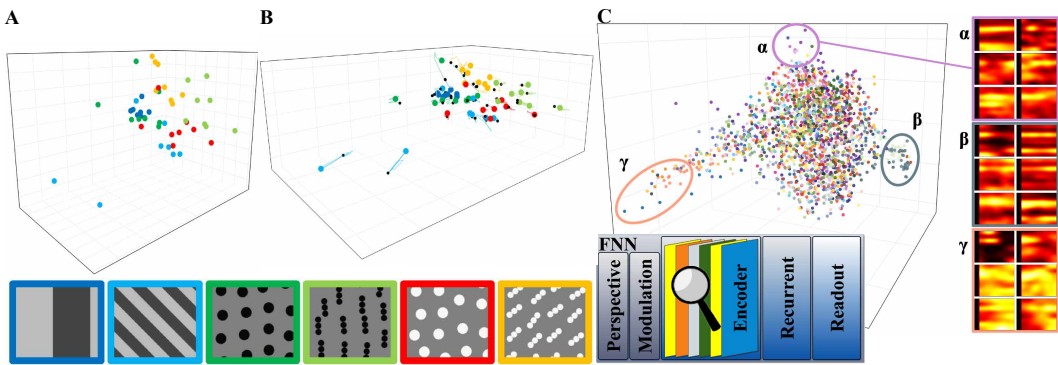

Figure 16: **Encoder L8 decoding manifold, trajectories and encoding manifold without intensity artifacts**. Without the intensity artifacts there is no temporal development at all in the decoding trajectories (comparable to encoder L1) apart from the jump after the 0-th step. The non-selective high intensity neurons are padding artifacts at the edges of the image. In the encoder, due to spatial convolutions, the effect of these artifacts spreads out across the feature maps. This is supported by the intensity smoothly organizing the manifold with a transition from intensity-only neurons to selective responses. In the recurrent stage, the function of the attention layer is capable of filtering exactly those artifacts out. The artifacts are reintroduced by the recurrent-output convolution, but then filtered out by the readout interpolation from central neurons only.

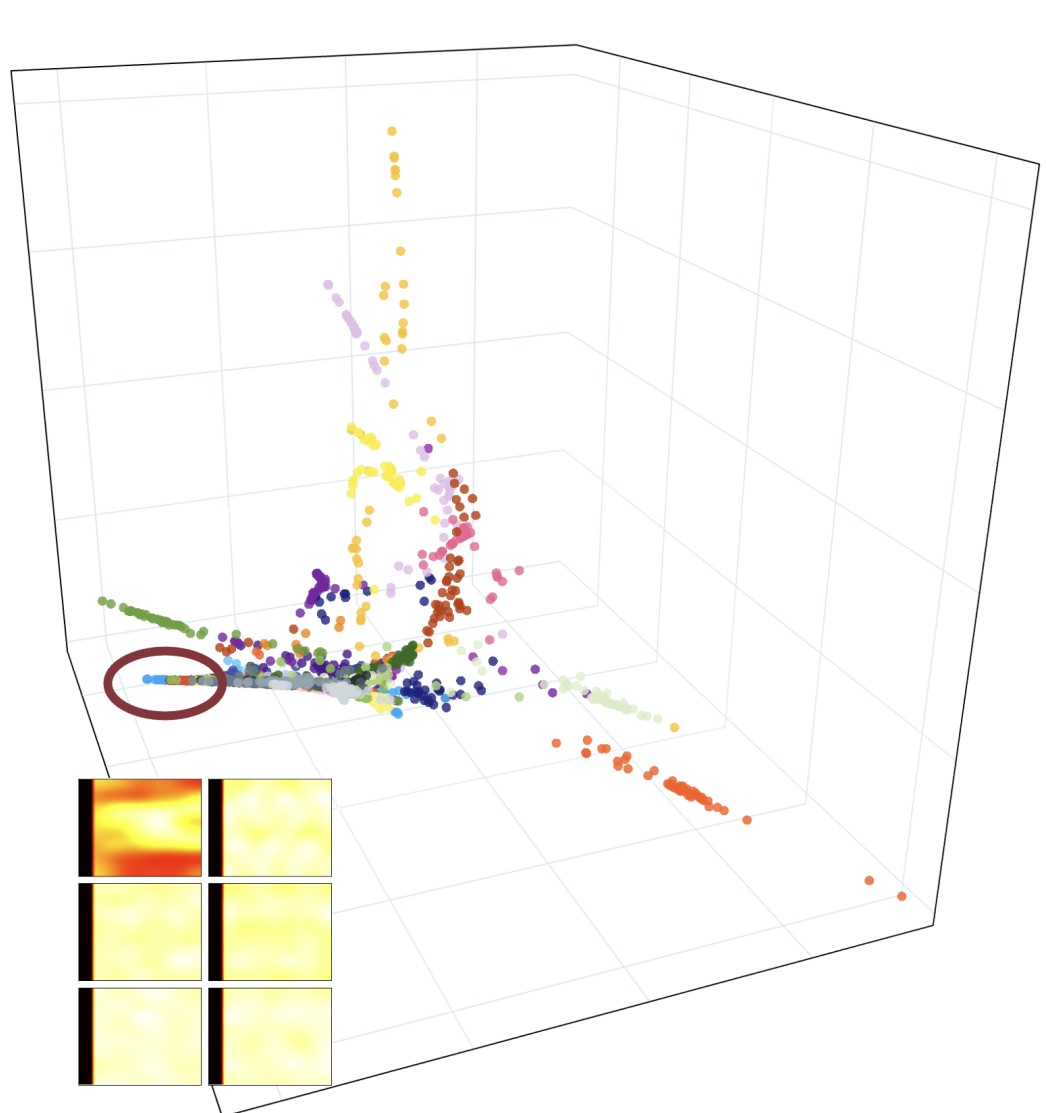

Figure 17: **Minimodel encoding manifold with intensity arm**. The intensity artifacts are also present in the border regions of feature maps in the model from Du et al. (2025).

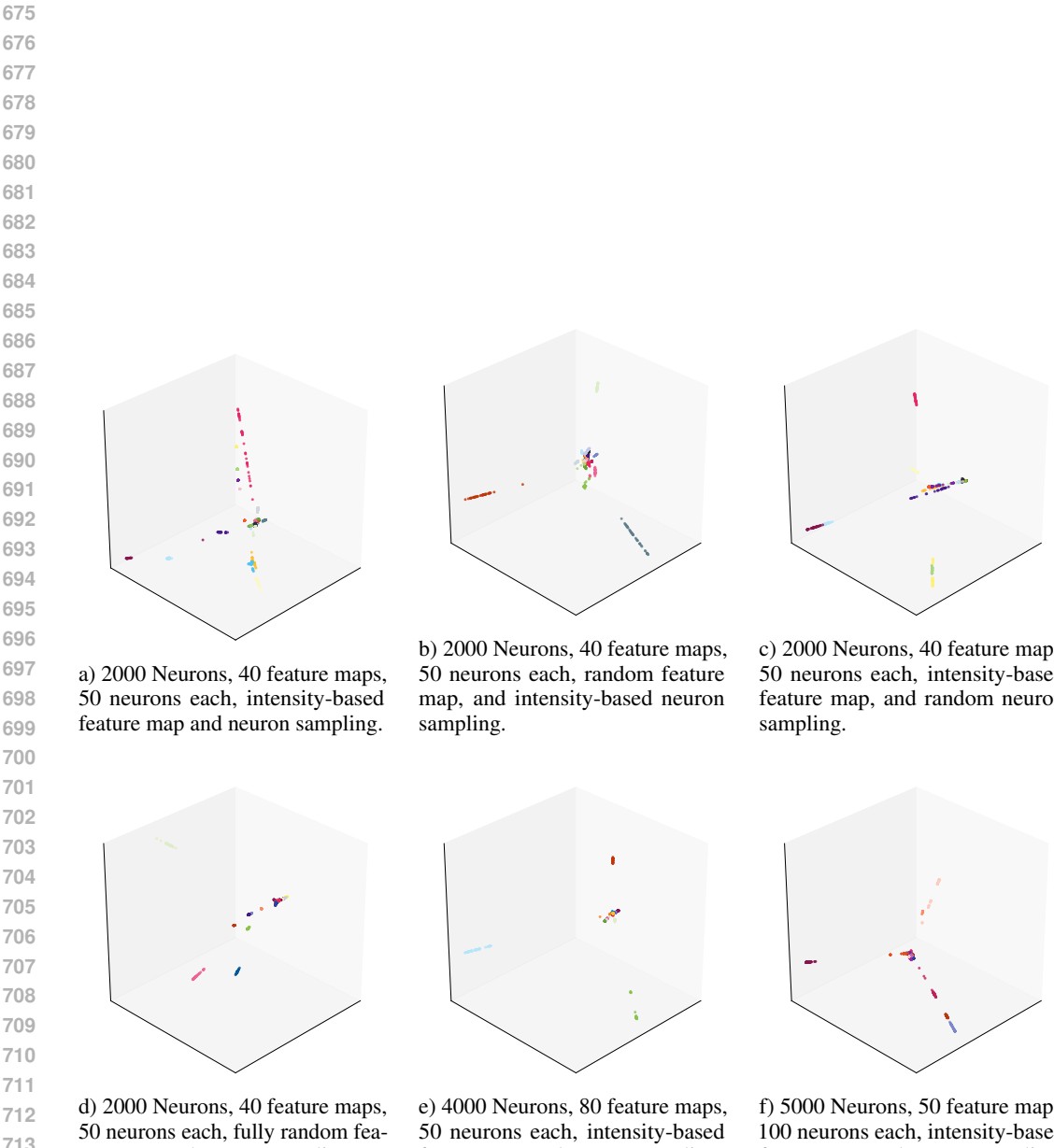

a) 2000 Neurons, 40 feature maps, 50 neurons each, intensity-based feature map and neuron sampling.

b) 2000 Neurons, 40 feature maps, 50 neurons each, random feature map, and intensity-based neuron sampling.

c) 2000 Neurons, 40 feature maps, 50 neurons each, intensity-based feature map, and random neuron sampling.

d) 2000 Neurons, 40 feature maps, 50 neurons each, fully random feature map and neuron sampling.

e) 4000 Neurons, 80 feature maps, 50 neurons each, intensity-based feature map and neuron sampling.

f) 5000 Neurons, 50 feature maps, 100 neurons each, intensity-based feature map and neuron sampling.

Figure 18: **Sampling tests for readout encoding manifolds**. The encoding manifold for all sampling conditions look similar, having clusters by feature maps.

