# OpenReview forum: "How Neural is a Neural Foundation Model?"
_ICLR.cc/2026/Conference — Submitted to ICLR 2026_

### Official Review · Reviewer_5PbF · 2025-10-15

**Soundness:** 2
**Presentation:** 1
**Contribution:** 2
**Rating:** 2
**Confidence:** 3

**Summary:**

The paper provides a novel technique to analyze representations from the intermediate layers of the neuronal foundational model from Wand et al 2025, suggesting that recurrent layers play a core role in the topology of internal representations.
They construct a neural encoding and neural decoding manifolds and try to capture the relation between two manifolds by the temporal evolution of neuronal activity.
They also introduce a tubularity metric to quantify the relationship between artificial and biological neural response trajectories.

**Strengths:**

1. The paper raises a very important question of interpreting neural foundational models, especially for biological plausibility, and do it in a novel and original way.
2. The paper uses state-of-the-art models and diagnostic tools, also using biologically interesting stimuli, etc
3. Careful statistical analysis is highly appreciated (Bonferroni corrections, etc)

**Weaknesses:**

1. **The paper is not clearly written**. The paper is not written in a self-sufficient manner. To understand and critically evaluate the central claims and contributions of the paper it is essential to understand the architecture of the Wang et al 2025 foundational model encoder. However, neither the original paper no the current paper provide a diagram / schematics of the encoder - are there skip connections? how many 3D convolutional blocks are there? Are they full 3D convolutions or are they factorized? What is the maximum temporal horizon, which could be seen by the temporal convolutions? Same applies for the tools they use for the analysis, the introduction even in the appendix is not enough for critical evaluation.
2.  **Lack of connection to the current interpretations of the data-driven networks**. The paper attempts to contribute towards interpretability of the neural networks, however, it seems like the authors have not exhaustively studies the current interpretations of the data-driven neural predictive networks.
Specifically, data-driven networks [1-3] with Gaussian readout [4] (Wang et al 2025 uses a variation of it) would interpret the encoder as  deconstructing visual stimuli into a set of leanrt basis functions and then the readout learns the position of per-neuron receptive field and a 1-dim vector. Under this interpretation, these 1-d vectors form the neural encoding manifold. This is not reflected anywhere in the text and I am not sure if the way the authors construct the neural encoding manifold is consistent with this perspective. This perspective and prior work this does not diminish the value of studying the intermediate representations in the encoder, but it might crucially change the interpretation of the results.
3. **The paper lacks clear explanations of technical details, which is crucial for interpretability studies**. See questions below

References:
[1] Klindt, David, et al. "Neural system identification for large populations separating “what” and “where”." Advances in neural information processing systems 30 (2017).
[2] Turishcheva, Polina, et al. "Reproducibility of predictive networks for mouse visual cortex." Advances in Neural Information Processing Systems 37 (2024): 7930-7956.
[3] Nellen, Nina S., et al. "Learning to cluster neuronal function."Advances in Neural Information Processing Systems 37 (2025)
[4] Lurz, Konstantin-Klemens, et al. "Generalization in data-driven models of primary visual cortex." ICLR 2021

**Questions:**

1. See questions in Weakness 1
2.  Do I get it correctly that all of the analysis was done on a single session? (Appendix A2, line 835 *"We used the FNN from session 8, scan 5."*)
3. Wrt to Weakness 2 - In appendix A3, lines 851-852 say *"50 neurons were sampled from each feature map"* - how exactly do you do it? are things sampled independently from each feature map or is not then what exactly do you do and why? How exactly *"the sampling probabilities of feature maps and neurons were set to be proportional to their activation strength"* (lines 852-853)?
4. How exactly have you adapted the stimuli to have 37 frames (lines 846-848)? Did you repeated some frames, put some preliminary buffer or just changed the shape?
5. In lines 162-163 you say *"We compare with biological decoding trajectories using the experimental data from Dyballa et al. (2024a)."*. Have you also finetuned the network from Wang et al for the new neurons? if yes - how (e.g. which components have been frozen if any)?
6. In which space (and dimensionality) tubularity is measured? time, amplitude of responses and repeats? Or is is the latent space after PCA transforms?
7. What do you mean by *"The encoding manifold for the readout layer is highly disconnected (Figure 4A), with each cluster corresponding almost exclusively to neurons sampled from a single feature map."* in lines 371-373? If this is about learning the receptive field  (*"* $p^n \in R^2$  *denote the spatial position (x, y)"* see Wang et al 2025 Methods section, *"Readout module"* paragraph) or do you mean a single dimension along the  $w^n \in R^C$  that *"denotes the feature weights for that neuron"*?
If first, than the clusters you report in *"Figure 4A"* under the papers from weakness 2 would be interpreted as grouping by receptive field rather than "cell subtypes".
8. What are the limitations of tubularity metric that you have introduced?

---

> ### Author Response · Authors · 2025-11-22
>
> We thank the reviewer for the detailed and insightful comments. We appreciate the assessment of our research as important and original, as well as the positive evaluation of our choice of models and tools, and statistical rigor.  We have addressed your major concerns regarding clarity, motivation, and quantitative results in our General Response above.
>
> __Specific Questions__
>
> __1. Methods and Encoder Details:__ We have added a figure on the encoding pipeline (__Figure 2__) and FNN architecture (__Figure 6__).
>
> __2. Single Scan:__ Results in the paper were obtained from a single scan, but we have validated our results using different scans. Note that this only affects the readout and output, since the core is trained on all scans. We briefly address this in the revised __Methods__.
>
> __3. Sampling:__ We have clarified the sampling procedure (see revised __Methods Appendix__).
>
> __4. Inputs:__ We have clarified how we obtain the 37 frames using the original script by Dyballa et al. (2024a); see revised __Methods Appendix__.
>
> __5. Finetuning:__ We have not done any finetuning, but argue that our analysis on the population level should compare across mice. We briefly address this in the revised __Methods__.
>
> __6. Tubularity Space:__ We have now clarified this: the calculations are performed in 10-dimensional PCA space (see __Methods Appendix__). We also invite the reviewer to see the additional comments posted to other reviewers about this.
>
> __7. Readout Interpretation (Weakness 2):__ The suggested background literature is appreciated! Since we want to investigate the representations of the readout, each point in the readout encoding manifold corresponds to a single neuron of the readout. The readout representations at one step have the shape n_biological_neurons=9000 x n_features=512, we sample 50 biological neurons from 40 features = 2000 readout neurons. The encoding manifold of biological neurons then is the output manifold. This approach ensures comparability to the other encoding manifolds. Our key message here is that those 512 features fulfilling very distinct functions (as the encoding manifold shows) are non-biological. We added this background to the revised __Prior Work__.
>
> __8. Tubularity Limitations:__ We have expanded on the limitations of the tubularity metrics (see revised __Limitations__) and added some grounding for them (see revised __Methods__).
>
> Thank you very much again for your review. Please do not hesitate to reach out in case you have any remaining questions or concerns.

---

> > ### Comment · Reviewer_5PbF · 2025-11-23
> > **More clarifications**
> >
> > Thanks a lot for improving the clarity.
> > 1. For Fig 6 with the FNN architecture - the non-linearities are skipped - are these always ReLUs apart from the last layer? Adding this detail in the caption would be sufficient. I believe the types of non-linearities are important to contentexualize your claims later
> > 2. thanks for clarifying
> > 3. Do you mean appendix A4?
> > 4. Which specific appendix do you mean? Could you clarify the page / number?
> > 5. Does the model have a shared perspective module for all sessions or is it a per-sesion module? if later, which one do you use for the new mice then? Also, perspective module is supposed to do the visual field transformations, which could substantially differ if the screen were located at different distances in the original Wang et al 2025 experiments and used Dyballa et al. (2024a) experiments. Could you please comment on it?
> > 6. Do you mean appendix A8? Why did you choose 10 dimensions and how does your metric changes behaviour with the change of this space dimensionality? Do the results qualitatively stay the same?
> > 7. I understand the readout is conceptually a feature-vector per neuron (e.g. 512 dimensional feature vector per biological neuron). I guess you call the whole per-mouse block as a readout, which clears some misunderstandings. However, why and how do you sample individual weights (feactures) from these 512 dimensional vectors?
> > Also, in the text you now have
> > *"For comparability, we use our encoding method to compare the embeddings of biological neurons and individual readout neurons, investigating not only the final embedding but also the readout embedding."* - what do you mean by the *"final embedding"* here?
> > 8. Thanks!

---

> > > ### Author Response · Authors · 2025-11-23
> > >
> > > Thank you for your quick answer and additional questions. We very much appreciate your detailed comments.
> > >
> > > 1. The FNN uses GeLUs, will add this.
> > > 3. Yes
> > > 4. A.3, essentially the stimuli were not static videos but Dyballa et al. 2024 provide a adaptive pipeline for generating stimulus videos for variable lengths.
> > > 5. The perspective (and also modulation) modules are used to model behavioral parameters of the mice (gaze, running speed, etc) during training. For inference, these are set to default values determined by Wang et al. 2025. The distance to the screen was 25cm in Dyballa et al. 2024, and 15 cm in Wang et al. 2025. We have adjusted stimulus sizes for comparable spatial frequency based on the stimulus expansion of the visual field. We highly value your awareness of these input details (alongside point 4) and can expand on them in the appendix.
> > > 6. Yes, we chose 10 dimensions to speed up computations. While absolute values change with different dimension, comparisons remain stable.
> > > 7. We do not investigate the weights used to combine the 512 feature vectors to the final output activity, but the individual feature activations. The sampling procedure is the same as for other layers where we interpret the 512 features as channels as before. This is also what we mean the the sentence concerning the final embedding: We investigate the readout neuron's encoding manifolds before combining the 512 features, showing the features fullfill very distinct functions (readout encoding manifold) and the final output encoding manifold (after combining the 512 features for each neuron). We will clarify that by final embedding we mean output encoding manifold.
> > >
> > > We hope this answers your questions, if not do not hesitate to ask for further details.

---

> > > > ### Comment · Reviewer_5PbF · 2025-11-23
> > > >
> > > > *Q4* I still did not get if there was a perspective module per session in the original Wang et al. 2025 model or if it was shared as the core is. Could you please clarify it?
> > > > *Q5* could i find this qualitative comparisons anywhere in the manuscript ?
> > > > *Q6* And **why** do you subsample 40 activations from the 512 dim latent space? and how do you interpret it?

---

> ### Author Response · Authors · 2025-11-26
>
> Thank you again for your quick reply.
>
> Q4: The perspective was trained individually, similar to the readout. Note, however, that the perspective module only outputs a distorted version of the input video based on mouse fixations (where data was available). It does not contribute to providing any representations of the model beyond the input. Moreover, if this data (mouse fixations, etc.) is unavailable, the perspective was also intended to be used with the default parameters of Wang et al. 2025.
>
> Q6: Similar to the investigation of other layers, we use this approach of subsampling to visualize data in the encoding manifolds. The exact number of samples was chosen to not hinder the visualization by showing too many feature maps, while still displaying the feature maps that responded to our stimuli. We validated this sampling procedure as described in the paper, but since there were additional questions about the sampling, we will add some results to the appendix.
>
> We again thank you for your concern with the details. We will upload a new version later that includes this, the justification for Q5, and other issues you mentioned in previous comments.

---

### Official Review · Reviewer_h5wS · 2025-10-28

**Soundness:** 2
**Presentation:** 3
**Contribution:** 2
**Rating:** 4
**Confidence:** 3

**Summary:**

The paper investigates the biological plausibility of a foundation neural network (FNN) trained to predict mouse visual cortex activity. The authors analyze the model’s internal representations using three neuroscience-inspired approaches: decoding manifolds, encoding manifolds, and temporal trajectory analysis. They introduce a new geometric metric tubularity to quantify the organization of neural trajectories and compare artificial versus biological neural dynamics. Results show that the FNN’s recurrent module introduces biologically plausible temporal dynamics, while feedforward and readout modules deviate significantly from biological organization. The authors argue for architectural adjustments (e.g., early recurrence, fewer redundant readout maps) to improve biological alignment.

**Strengths:**

1. The idea of probing foundation models through a neuroscience lens is both original and timely, offering a valuable bridge between computational neuroscience and modern AI.

2. The proposed tubularity metric is a creative and potentially useful approach for quantifying the geometric structure of neural trajectories.

**Weaknesses:**

1. The experimental analysis lacks ablation or sensitivity studies. It remains unclear how stable the conclusions are with respect to hyperparameter settings or architectural choices.

2. The discussion of model mechanisms is descriptive rather than mechanistic. Related work such as “What Has a Foundation Model Found? Using Inductive Bias to Probe for World Models” offers a more fundamental exploration of inductive biases.

3. The validation of the “tubularity” metric is limited. There is little theoretical grounding, and no comparison with alternative geometric or dynamical measures.

4. Figure 1 could be more informative and visually integrative. A clearer schematic that unifies the core workflow (encoding/decoding manifolds, tubularity computation, and main findings) would greatly improve readability and help the reader grasp the conceptual flow at a glance.

**Questions:**

1: On the core premise:

a. The paper equates tubularity with biological plausibility, but the argument relies almost entirely on the geometry of encoding/decoding manifolds and trajectories. Could these geometric measures alone truly capture brain-like computation?

2: On the definition and robustness of “tubularity”:
a. How is tubularity theoretically grounded? Could concepts from topology or knot theory (e.g., linking number, curvature-based tightness) provide a more rigorous mathematical basis?

b. Can properties or invariants be derived analytically from the current metrics to demonstrate robustness or internal consistency?

c. How sensitive are the proposed metrics to hyperparameter settings—for example, clustering thresholds, quantile, or binning strategy?

d. Would a bin-wise normalization approach yield a more stable, scale-free comparison across models or datasets?

e. Have you considered alternative distance measures, such as Dynamic Time Warping (DTW), for comparing trajectory similarity? What advantages or disadvantages does your approach have?

f. Conceptually, how do we know that higher tubularity truly correlates with better computational function?


3: On model design and experimental interpretation:

a.  The encoder’s inability to separate stimuli seems expected, as early layers mainly act as low-level feature extractors. In biological vision, early retinal processing (e.g., from photoreceptors to bipolar cells) lacks strong recurrent connections; each stage performs relatively modular, feedforward computation. Given this biological organization, why does the paper suggest introducing recurrence into earlier encoder layers as a way to make the FNN more biologically plausible?

b. The “intensity arm” is attributed to convolutional padding artifacts, but might this reflect a deeper architectural discrepancy between CNNs and biological vision—where visual space is continuous and boundary-free? Is “fixing padding” merely treating the symptom rather than the cause?

c. The paper reuses parameters (e.g., tensor decomposition rank) from Dyballa et al. (2024a) without justification. Given the difference between sparse, discontinuous biological responses and continuous activations in FNNs, could this distort manifold topology?

d. The neuron sampling strategy prioritizes highly active units (2000 neurons total), excluding low-response neurons that may play inhibitory or regulatory roles. Could this bias the manifold geometry toward excitatory-like representations?

e. Given that the recurrent module is preceded by an attention layer, how can we disentangle their respective contributions? Without ablation or control experiments, can we confidently attribute the observed improvements solely to recurrence?

4: Given the conceptual density of the paper, a comprehensive schematic figure that consolidates the key elements, such as model architecture, experimental pipeline (encoding/decoding manifolds, trajectory analysis), definition of tubularity, and the main conclusions, would greatly enhance accessibility. This could replace or extend the current Figure 1 to serve as a visual summary of the study’s logic and findings.

---

> ### Author Response · Authors · 2025-11-22
>
> We thank the reviewer for the detailed and insightful comments. We appreciate the assessment of our approach as valuable, timely, and original, as well as the usefulness of our proposed tubularity metric. We have addressed the major concerns regarding clarity, motivation, and quantitative results in our General Response above.
>
> Specific Questions
>
> __1.__ Tubularity alone does not capture biological plausibility. Rather, it is a measure of neural activity that is observed experimentally; it is from the biological data that the motivation for tubularity arose. We exploit it empirically, by asking whether the FNN dynamics resemble – coarsely – the biological dynamics. It provides an additional data point to the already established metrics now included in the paper. Nevertheless, we believe it is a striking feature of the data.
>
> __2.__ We acknowledge that tubularity analysis is preliminary and needs further investigation in the Limitations.   And we agree that the analysis is only beginning and appreciate your comments about knot theory, linking numbers, etc. We are very much on the same “page” as you, but have been pursuing somewhat different directions. In particular, we have been working with properties of dynamical systems to build a more formal theory. Our hope is to extend it,  and to show how it specifies the exponential map to the encoding manifold. Loosely speaking, neurons whose tubular neighborhoods overlap are “close” in the encoding manifold, and their “inverses” relate to the decoding manifold. And multi-selectivity of neurons may be related to the transverse crossings of the trajectories. But as (we hope) you will agree, this analysis would take us quite afar from the main story of this paper, which is how FNN relates to the mouse visual system. We hope to publish these additional theoretical analyses in the near future.
>
> b) Thank you for raising this point – we may need a bit more clarification to make sure we address your concern correctly. Could you please specify which aspect of consistency or robustness you’re referring to? All of our computations were performed on discretized data, with bin sizes and trajectory scales matched to the temporal resolution of the neural spike trains. The trajectory bundles were defined to include 90% of the data, and crossings were evaluated at this same scale. We just want to confirm that this is the kind of robustness issue you had in mind.
>
> c) Preliminary tests showed robustness to small variations in these hyperparameters. Since both the units in FNN and the neurons in the mouse were sampled, the issue of how to set these parameters is a study in itself. While we again acknowledge the preliminary nature of our approach, we stress that such tubularity metrics and their application in neural dynamics is just beginning. As we mentioned above, we are already working on building better foundations; and we hope that our paper will inspire others to do so as well. The integration of dynamics with manifold geometry and topology will be, we predict, an important line of investigation in future.
>
> (d) Regarding bin-wise normalization: we already compute the metrics using a binned representation of the trajectories, aligned with the temporal scale of neural activity. This approach effectively normalizes comparisons across models and datasets by working at a consistent resolution. We have added clarifying details on our current binning and normalization procedure in the __Methods Appendix__.
>
> e) We chose distance to the centerline as a simple way of measuring tightness. We did consider dynamic time warping, but discounted it because the time scale is defined by the stimulus presentation. Other spaces in which to study the tubular metric, in e.g. Hardy spaces (to include derivatives), and earth-movers distances were also considered but their analysis was too difficult at this point.
>
> f) We show that biological data is organized in these tubes. As stated above, this is an empirical observation. Other aspects of the tubular neighborhoods, e.g. how tightness varies with curvature, may well inform issues of neural dynamics. For example, if curvature derives from adaptation, one might expect tightness to remain constant, but if it arises from feedback, then it might enlarge. On the other hand, the rapidly expanding trajectories found for the FNN readout layer corresponded to the tightly clustered encoding manifold. This does not reflect biological reality, or feedback. Rather, it just resembles repulsion. In short, we agree that future work should investigate more in detail what the dynamics means computationally and comment above that we are also working on this.

---

> > ### Author Response · Authors · 2025-11-22
> >
> > __3.__ a) The late-stage encoder already holds quite some computational power in many layers, and finds more global features. The retina arrives at its final representations quickly,  and builds more local features. Further, the retina does include some temporal/recurrent mechanisms, such as via amacrine cells and gap junctions. Thus, we suggested including recurrence after a more light-weight local feature encoder to better align the model with biology.
> >
> > b) We agree that there is a difference, so the potential fix could be an attempt to also make the artificial network’s convolutions more boundary free, e.g. via applying circular padding.
> >
> > c) We emphasize that we did not reuse their hyperparameters, but only their published  procedure. We clarified this in the revised __Methods Appendix__.
> >
> > d) In the visual cortex, inhibitory neurons are actually not characterized by low responses but instead by high firing rates and low selectivity. See, e.g.:
> > Kerlin AM, Andermann ML, Berezovskii VK, Reid RC. Broadly tuned response properties of diverse inhibitory neuron subtypes in mouse visual cortex. Neuron. 2010 Sep 9;67(5):858-71.
> > Hofer SB et al., Differential connectivity and response dynamics of excitatory and inhibitory neurons in visual cortex. Nat Neurosci. 2011 Jul 17;14(8):1045-52.
> > The sampling procedure we adopted was used in Dyballa et al. (2024a) to capture neurons that actually care about the input stimuli in the manifolds and prevent non-responsive neurons from obscuring the analysis. We followed their procedures as closely as we could to enable comparison of the FNN with the published biological data. Importantly though, our experiments with other variations of the random sampling strategy did not qualitatively change the results; we now state this important technical check in the __Method Appendix__.
> >
> > e) The FNN was trained both with and without attention, and the authors in attention did not contribute to model performance (Wang et al., 2025). Therefore, the recurrence is likely the key driving factor. Since they only published the model with attention, we were constrained to using this for our analysis. We have now included this in the revised __Discussion__.
> >
> > __4. __Since significant parts of our method rely on visualizations, we believe that the newly restructured __Figure 1__ is more adequate for providing an overview of the paper by showing our analysis tools. We also restructured the results figures to provide a better overview and to convey the key points “at a glance.”
> >
> > Thank you again for your review. Please do not hesitate to reach out in case you have any remaining questions or concerns.

---

> ### Comment · Reviewer_h5wS · 2025-11-26
>
> We thank the authors for their detailed response. While several methodological clarifications are appreciated, the rebuttal does not satisfactorily address the fundamental conceptual issues raised in the original review.
>
> First, the authors acknowledge that tubularity is a “preliminary” and “empirical” measure whose theoretical grounding is still under development. This admission, although transparent, reinforces our core concern: it is methodologically problematic to elevate a geometric property that lacks rigorous definition, validation, and theoretical justification to the status of a gold-standard criterion for “biological plausibility.” The central conceptual questions remain unresolved:
> What computational advantage does tubularity confer (e.g., robustness, sample efficiency, inductive bias)?
> Is tubularity a necessary computational principle, or merely an incidental biological epiphenomenon?
> Without answers to these questions, enforcing alignment to this property amounts to imitation rather than principled model design.
>
> Second, regarding the metric’s properties (Question 2b), the justification remains insufficient. When introducing a novel metric, it is incumbent upon the authors to demonstrate its advantages over existing alternatives. The response that more sophisticated sequence-comparison tools (e.g., Dynamic Time Warping) were considered but deemed “too difficult” does not address the critical issue: why should a simple Euclidean-distance–based measure be expected to capture computationally meaningful structure in neural dynamics better than metrics explicitly designed for temporal alignment and comparison? The burden of proof rests with the authors to show that their chosen metric is not merely convenient, but is uniquely sensitive to the functionally relevant aspects of the phenomena under study.
>
> Finally, the paper continues to rely on the unexamined assumption that “more brain-like = better.” This premise remains unsubstantiated. If the goal is predictive performance, biological constraints may in fact degrade the model’s capabilities. Conversely, if the goal is scientific understanding, a model that performs well while diverging internally from biological patterns may be more revealing, by demonstrating alternative valid computational solutions rather than reinforcing an assumed optimality of biological dynamics.
>
> I have also reviewed the questions raised by the other reviewers, and I agree that they highlight valid weaknesses that need to be addressed.

---

> > ### Author Response · Authors · 2025-11-26
> >
> > We appreciate the reviewer’s concerns and have revised the manuscript to address and clarify them.
> >
> > __1. Tubularity is not presented as a gold-standard criterion for biological plausibility__
> >
> > We apologize if our framing suggested that tubularity was intended as a normative or “gold-standard” metric. That is not the intended claim. In the revised manuscript, tubularity is positioned explicitly as:
> > - an __observed statistical regularity__ in biological population dynamics,
> > - a geometry-level descriptor __complementary to existing alignment metrics__, and
> > - a tool for __identifying qualitative differences__ in temporal organization,__not as a principle that models must satisfy__.
> >
> > To prevent misunderstanding, we have added the following clarification in the Discussion:
> >
> > _"Tubularity was developed as a descriptive, data-driven characterization of population-level temporal organization. Rather than constituting an optimality principle for model design, it highlighted a salient structural property empirically present in biological recordings that was absent in early FNN layers."_
> >
> > We believe this resolves the concern about elevating tubularity beyond what the data support.
> >
> > [1] Berndt, Donald J., and James Clifford. "Using dynamic time warping to find patterns in time series." Proceedings of the 3rd international conference on knowledge discovery and data mining. 1994.
> >
> > [2] Alt, Helmut, and Michael Godau. "Computing the Fréchet distance between two polygonal curves." International Journal of Computational Geometry & Applications 5.01n02 (1995): 75-91.
> >
> > [3] Nutanong, Sarana, Edwin H. Jacox, and Hanan Samet. "An incremental Hausdorff distance calculation algorithm." Proceedings of the VLDB Endowment 4.8 (2011): 506-517.

---

> > > ### Author Response · Authors · 2025-11-26
> > >
> > > __2. Why introduce tubularity, and why this particular metric?__
> > >
> > > The purpose of tubularity is not to replace temporal-alignment methods such as DTW [1], but to __fill a gap__: existing temporal metrics compare _pairs_ of trajectories, whereas the empirical phenomenon we are testing concerns the organization of __entire trajectory bundles__. This bundle-level geometry is not captured by DTW. Other standard tools such as Fréchet [2] and Hausdorff [3] distances, for example, or RSA- or CCA-derived dynamics measures, compare trajectories pairwise and do not characterize collective geometric structure of a set of curves. __What we observed empirically in biological data (visible directly in Figs. 5A and 5D) is that trajectories form coherent bundles with shared centerlines and structured crossings. Existing metrics do not provide a way to quantify such population-level geometric features.__
> > >
> > > We have now added the following to the Methods section to make this explicit:
> > >
> > > _“...tubularity is not a trajectory-matching metric but a population-geometry metric: it assesses the structure of collections of trajectories rather than the similarity of individual pairs."_
> > >
> > > Although our use of tubularity is empirical, the concept is standard in differential geometry: if we assume dynamics live on some embedded manifold, the tubular neighborhood is a reasonable object to study [4, 5], and related work has used tubular neighborhoods to characterize well-behaved dynamical trajectories (e.g., [6]). These results provide conceptual grounding for why tube-like organization is a natural population-level feature to quantify.
> > >
> > > Therefore, our use of Euclidean geometry is not a convenience assumption, but a natural starting point for analyzing population-level structures: (i) it aligns well with how manifold-learning methods represent neural state space (e.g., diffusion maps, PCA); (ii) it is invariant to rotations, translations, and temporal rescaling; and (iii) it offers transparency and interpretability (satisfying the reviewer’s concerns regarding "rigorous theoretical justification").
> > >
> > > Finally, we do not claim that this metric is definitive; rather, it is the first quantitative assessment of a property found in biological recordings but missing in early model stages.
> > >
> > > We have strengthened this perspective in the revision highlighted tubularity as a descriptive tool, not a requirement:
> > >
> > > _"Tubularity was developed as a descriptive, data-driven characterization of population-level temporal organization. Rather than constituting an optimality principle for model design, it highlighted a salient structural property empirically present in biological recordings that was absent in early FNN layers."_
> > >
> > > We believe this clarifies both why the metric is needed and how it is used: as a descriptive tool for diagnosing population-level structure rather than an imposed biological standard.
> > >
> > > [4] Ana Cannas da Silva. Lectures on Symplectic Geometry, vol. 1764, Lecture Notes in Mathematics. Springer Berlin Heidelberg, Berlin, Heidelberg, 2008.
> > > [5] https://ncatlab.org/nlab/show/tubular+neighborhood+theorem
> > > [6] Kim, Chanwoo, and Donghyun Lee. "Decay of the Boltzmann equation with the specular boundary condition in non-convex cylindrical domains." Archive for Rational Mechanics and Analysis 230.1 (2018): 49-123.

---

> > > > ### Author Response · Authors · 2025-11-26
> > > >
> > > > __3. More brain-like is not necessarily better__
> > > >
> > > > We fully agree with the reviewer that “more brain-like” internal structure is not inherently better. As we clarified in response to reviewer __MUj9__, our focus is narrower and tied to the specific scientific use-case of the FNN as "an accurate functional digital twin of the mouse visual system", cf. Wang et al. (2025) [7]. In such settings, the goal is not only prediction but also interventions, counterfactuals, and mechanistic reasoning. For these, the internal computations do matter. A model can match outputs but fail under perturbations or OOD conditions that are routinely used in neuroscience experiments, such as optogenetic silencing, pharmacology, etc. Our manifold analyses are intended precisely for this "look inside the box" requirement, namely to reveal when the latent state space supports (or not) the same separations and temporal structures that biological circuits rely on.
> > > >
> > > > Thus, we do not assume that “more brain-like = better” in general; but for digital-twin models, internal alignment is desired for reliable mechanistic interpretation. Our revision mentions this motivation already in the Intro:
> > > >
> > > > _"while the FNN learned a forward map reasonably well, it processes stimuli quite differently from the mouse, and hence is only a partial digital twin in the dynamical sense. Importantly, our manifolds identify where the disparities lie."_
> > > >
> > > > We appreciate the reviewer's point and have now also added the following passage to the Conclusion, in order to address this explicitly:
> > > >
> > > > _"While biological fidelity is not a prerequisite for achieving high predictive accuracy, digital-twin use cases require enough internal alignment to support mechanistic and interventional inference."_
> > > >
> > > > [7] Wang, Eric Y., et al. "Foundation model of neural activity predicts response to new stimulus types." Nature 640.8058 (2025): 470-477.

---

### Official Review · Reviewer_ppwn · 2025-10-30

**Soundness:** 2
**Presentation:** 3
**Contribution:** 1
**Rating:** 2
**Confidence:** 3

**Summary:**

This paper compares a recently published foundation model of neural responses (FNN) in mouse visual areas (Wang 2025) to data from mouse retina and V1 (Dyballa 2024) recorded while viewing different visual flows. They use decoding and encoding manifolds to visualize high-dimensional activations (possibly across time) and feature selectivity in both systems then evaluate the extent to which the FNN aligns with the biological visual system. To make these comparisons more concrete the authors introduce “tubularity” which roughly measures the tightness of different temporal trajectories in activation space. They conclude that the recurrent module of the FNN produces responses which best align with recorded neural data. Most significantly, this is the first layer where they found robust decodability of stimulus type. Decodability corresponded to tighter trajectories according to their tubularity measure, though tubularity in the FNN recurrent module was still “sub-biological.” By contrast the activity in the FNN readout module was a poor match for biological data.

**Strengths:**

The strongest point of this paper is establishing that the very intuitive correspondence from feedforward encoder to retina and recurrent module to V1 doesn’t actually hold. It is well written and the subject matter is timely given the proliferation of foundation models across the field.

**Weaknesses:**

I think after reading I now know where in the FNN representations qualitatively resemble the mouse visual system - the recurrent module and not encoder stage - but I’m not sure why this is the case, how encoder computations support this similarity in the recurrent module, whether or which parts of the recurrent module is more similar to V1 or retina etc. Given the lack of deeper insights here the significance and impact to the community isn’t obvious and I’m leaning reject.

1. The biggest weakness of the paper is that there is very little analysis that follows up the surprising mismatch between FNN modules and mouse visual system. Given that there is now a very large literature on mapping large artificial networks to brain data I think something like this would have vastly improved the paper. For example, take the fact that only the recurrent module shows the tubularity and decodability present in retina and V1 data. A natural follow up (I think within the stated goals of the paper) would be to ask if there are parts of the recurrent module that fit retinal data better than V1 and vice versa (via regression of FNN activations onto neural data and an examination of the regression coefficients). Also along these lines, it would be interesting to know if there are subpopulations in the recurrent module that are more sensitive to the feedforward input (this could be checked simply by plotting the distribution of input weights to the recurrent module). Can you fit retinal data well just with these subpopulations? Some more analyses along these lines would go towards actually explaining how the FNN and biological visual areas process inputs.

2. I didn’t find that the measure of tubularity was very well motivated. Since it was the main novel method introduced in the paper I think a comparison with other metrics would’ve significantly strengthened the paper. For example, does tubularity allow us to conclude anything beyond what the simple decoding analysis revealed? I take the author's point that measuring tubularity shows that though recurrent modules can decode stimulus type, the trajectories are less “tight” than biological trajectories. But again, this conclusion suffers from a lack of further analyses which show that this is an important difference from the perspective of computation and representation. Why should we care about relative tightness if across the entire noisy data set both systems can linearly separate stimulus type?

3. I was a bit confused about why the readout was included in the analyses. If my understanding is correct, the authors are analyzing a readout module from the original Wang paper, session 8 scan 5. The weights of this module are trained to predict data from an individual mouse. They then compare this module with recordings from a completely different animal. Given how the training of the FNN is set up one would very strongly expect there to be no correspondence. Could the author clarify why we might expect a correspondence given that this part of the system was trained exclusively to predict activity from cells that aren't present the dataset examined?

4. I found some of the recommendations a bit speculative and confusing. The authors suggest introducing recurrence earlier on to make activations more biologically plausible, yet point out elsewhere that the retina is itself not strongly recurrent. There’s a tension here between introducing biologically non-plausible circuit mechanisms to introduce more biological plausible activation patterns that I think deserved a bit of attention.

**Questions:**

See above section.

---

> ### Author Response · Authors · 2025-11-22
>
> We thank the reviewer for the detailed and insightful comments. We appreciate the recognition of the importance of our research topic and of the relevance of our approach. We have addressed the major concerns regarding __clarity, motivation, and quantitative results__ in our __General Response__ above.
>
> __Specific Questions__
> - __Subpopulation Analysis & Mechanism:__ We followed the suggestion of fitting biological data from the recurrent activations (and other layers) (see the heavily revised Section 3.4). However, although the specific subpopulation analysis proposed (isolating neurons sensitive to feedforward input) would be highly interesting, it is technically infeasible because the encoder input and hidden states are mathematically mixed in the Convolutional LSTM block before processing (see the new Figure 6 in the Appendix). Consequently, weights cannot be cleanly disentangled to identify a "retinal-like" subpopulation.
> - __Tubularity & Tightness:__ We have updated the text to better motivate the notion of tubularity (see __Methods__). While a noisy cloud of points can be linearly separated (as in the FNN), the biological system still structures this activity into tubular trajectories. Our tubularity metrics are an attempt to quantify some of these aspects beyond linear classification. As we stated above: Tubularity specifically quantifies the _stimulus-dependent_ emergence of temporal trajectories, distinguishing biologically plausible signal development from simple artifactual smoothness. Moreover, the curvature of the trajectories is important: we conjecture that it relates to feedback signaling and neural adaptation. Moreover, tubularity opens the connection between dynamics and the encoding arrangement: those neurons that share tight tubular neighborhoods are close in the encoding manifold.  We acknowledge this is preliminary and needs further investigation in the __Limitations__ section.
> - __Readout Validity:__ We argue that analyzing the readout is valid because we are comparing __population-level functions__, not individual neuron tuning. While the specific weights are fit to one animal, the _computational function_ of the readout generalizes. We have further validated these findings across different scans to ensure robustness. We now briefly address this in the __Methods__ but the results were so similar that we didn’t feel it necessary to include all of them. They could easily be added, if you think that it would be worthwhile.
> - __Recurrence & Retina:__ We have clarified the discussion regarding retinal recurrence. Biological retinas do utilize lateral and recurrent connections (e.g., via amacrine cells and gap junctions between retinal ganglion cells) to shape temporal dynamics. Furthermore, there is substantial adaptation in the retina. Our suggestion, therefore, is to replace the deep, static feedforward encoder with a shallower, recurrent architecture to better mimic this biological compression of spatiotemporal features. We invite you to see our additional comments to reviewer __pp5o__.
>
> We thank you again for your review. Please do not hesitate to reach out in case you have any remaining questions or concerns.

---

> > ### Comment · Reviewer_ppwn · 2025-11-26
> > **Response to comments**
> >
> > Many thanks to the authors for the thoughtful consideration of my questions and suggestions. In particular, I found the explanation of tubularity much clearer. I think the issues I still have are best summarised by this statement in the response "Moreover, the curvature of the trajectories is important: we conjecture that it relates to feedback signalling and neural adaptation." To give a full acceptance score I think this would've had to go beyond conjecture. If the author could establish this then this would be a very strong paper indeed. Without this however, I think it remains unclear whether tubularity really captures something essential about how the brain is processing information, and therefore whether alignment between FNN and data according to this metric is something of interest to the community. Given this I'll raise my score to a 4.

---

### Official Review · Reviewer_pp5o · 2025-11-01

**Soundness:** 3
**Presentation:** 1
**Contribution:** 3
**Rating:** 6
**Confidence:** 3

**Summary:**

The authors use a computational neuroscientist's toolbox to analyse the FNN, a foundational model of the mouse visual cortex

Namely, they find so-called decoding and encoding manifolds using the kinds of simple stimulus videos one would expect to see in neuroscience experiments, and characterise them with a novel 'tubularity' metric.

They find that the feedforward encoder lacks biologically plausible stimulus-dependent temporal patterns despite having temporal convolutions, and that the recurrent module learns seperable stimulus representations. However, V1 trajectories are shown to be more tubular than those developed by FNNs.

**Strengths:**

This is a good, principled approach to understanding foundational models, and certainly the correct flavour of research to do with them now that we have obtained them. The paper goes on to provide concrete architectural suggestions (early-stage recurrence, feature dimensionality constraints). Similarly, the identification of padding artifacts as a source of non-biological behavior is a valuable practical finding.

**Weaknesses:**

Figures don't do the approach justice, and more time spent on styling would make the paper much more visually pleasing to read. e.g. PSTH were too small/low resolution to really inspect.

Tubularity is not well motivated - it is mentioned as a similarity mteric for biological and artifical systems, but the reasoning for this specific form is not provided. e.g., why not just use straightness of manifolds? Overall, there was a lack of comparison to the wider range of measures to characterise neural trajectories.

Wording was difficult to parse in multiple places, namely when discussing the encoding manifold 'arms' on page 5.

**Questions:**

Why only analyze L1 and L13 from the 15-layer encoder? A more systematic analysis across all layers would reveal how representations gradually develop. What guided this choice?

The tubularity metric is novel and mathematically defined, but lacks comparison to alternative measures. Have you considered simpler metrics like trajectory straightness, smoothness, or variance?

You suggest adding recurrence to early encoder stages - can you be more specific about how this could be achieved?

---

> ### Author Response · Authors · 2025-11-22
>
> We thank the reviewer for the detailed and insightful comments. We appreciate the positive assessment of our research approach, and the recognition of the importance of our practical findings. We have addressed the major concerns regarding __clarity, motivation, and quantitative results__ in our __General Response__ above. In particular, we strongly agree that the figures needed work, and they have essentially been redone.
>
> __Specific Questions__
> - __Tubularity:__ We have expanded the text to better motivate Tubularity. Standard metrics like "straightness" or "smoothness" measure geometric properties independent of the driving signal. In contrast, Tubularity specifically quantifies the _stimulus-dependent_ emergence of temporal trajectories, distinguishing biologically-plausible signal development from simple artifactual smoothness. Moreover, the curvature of the trajectories is important: we conjecture that it relates to feedback signaling and neural adaptation. Tubularity opens the connection between dynamics and the encoding arrangement: those neurons that share tight tubular neighborhoods are close in the encoding manifold.  We acknowledge this is just the beginning of such analyses, and remark that further investigation is needed in the __Limitations__ section.
> - __Layer Selection:__ To better cover the full encoder stage, we have added the necessary quantitative results and manifold visualizations for intermediate layers to the Appendix. One detail is that we restricted our analysis to layers immediately following activation functions to ensure that we sampled the non-linear feature space of the model following the literature; e.g. Dyballa et al. (2024a).
> - __Early Recurrence:__ By "adding recurrence to early stages," we propose replacing the deep feedforward encoder with a shallow, retina-like encoder (extracting local spatiotemporal features) that feeds directly into a recurrent core. This would mimic the biological visual hierarchy, where recurrence arises at the later stages in the retina, with amacrine and RGC gap junction connections, rather than after a deep feedforward stack. And it is important to keep the granularity of the stage analogous to the retina in proportion to that representing the cortex.
>
> Thank you again for your review. Please do not hesitate to reach out in case you have any remaining questions or concerns.

---

### Official Review · Reviewer_MUj9 · 2025-11-02

**Soundness:** 2
**Presentation:** 1
**Contribution:** 2
**Rating:** 2
**Confidence:** 4

**Summary:**

This paper aims to study the internal representations of a specific, recently proposed digital-twin–style neuro-foundation model that predicts neural activity responses to visual stimuli, in an attempt to understand the computations in such models, and also whether the computations performed by this model or its learned representations resemble those of the brain itself. The authors approach this question by looking at neural encoding and decoding manifolds, where they analyse for several layers or modules in the model, the similarity in encoding properties of neurons across stimuli (using the neuron factor from tensor decompositions to indicate shared or similar circuit properties) and trial-wise analysis of neural population dynamics (condition-averaged low-dimensional projections of neural activity trajectories) respectively. The authors also introduce a metric, tubularity, through which they quantitatively compare trajectories extracted from the model and trajectories in neural state space (mouse neural activity). Overall, the authors find that the earlier encoder components of the model do not show much similarity to biological networks, but that recurrence and additional constraints on output modules could starkly improve the bio-plausibility of these foundation models.

**Strengths:**

* I think the paper is focused on an important endeavour. Interpretability is an important research direction – so we have an understanding of how models arrive at their predictions – and adopting a dynamical systems approach inspired by computational neuroscience seems like the most promising candidate to me for obtaining useful insights into model function.
* The authors have introduced a metric, tubularity, which can be compared across contexts (i.e., artificial neural trajectories vs. biological ones), which directly gets at the question of comparing models to brains in a quantitative way.

**Weaknesses:**

* One of the main issues I have with this paper is its lack of clarity and poor presentation. The figures are nowhere near publication-ready and it is very hard to really understand what's going on from them – which severely impacts my rating as the main focus of the paper is interpretability. I would highly encourage the authors to take inspiration from other computational neuroscience papers and spruce up their figures, and also improve the writing:
    * The introduction lacks a clarity, doesn't have a clear flow or story, and does not do a good job of motivating the study. I was left asking, "why should a foundation model that is a strong predictive model of neural activity actually be brain-like?" – maybe it doesn't have to be, and the authors are simply curious whether the representations are akin to those of the brain, but this wasn't made clear. Furthermore, what does it mean when the authors say that neuro-foundation models "generalise" to OoD scenarios, especially in reference to Fig. 1A which seems unclear? Similarly, what does it mean for them to be "supportive" or "asking questions"?
    * Some acronyms, e.g., FNN, PCA, are never defined fully prior to using the acronym. In general, several paragraphs explain results with jargon and assume familiarity on the reader's part with the techniques and how to interpret them. I found it quite hard to understand what exactly was being conveyed, e.g., in Section 3 – and some of the claims did not seem very apparent to me from the figures. Furthermore, details on the FNN model being considered are quite sparse, both in the main text and appendix – this should be described better as it is central to the paper.
    * Overall, I wasn't able to clearly evaluate whether the claims were supported by clear, unambiguous evidence due to a combination of the figures and writing (see also my point below on lack of quantitative analyses).
* Another key issue with the paper is the lack of quantitative analyses apart from the one tubularity-based comparison. There is no attempt to compare the dynamics of the FNN's trajectories to those of actual neural activity based on metrics such as Dynamical Similarity Analysis (DSA) or Canonical Correlation Analysis (CCA). These have been used in many works in the literature to compare the dynamics and representational geometry of models and actual neural activity, e.g., https://www.biorxiv.org/content/10.1101/2024.10.04.616712v3 and https://www.biorxiv.org/content/10.1101/2025.02.07.637062v1. Without comprehensive quantitative analysis, unfortunately interpretability becomes akin to reading tea leaves based on visualisations, and is quite far-removed from the real data. As we are looking at reduced-dimensional spaces extracted using specific methods, we are limited by the assumptions or flaws of those methods.
* Only specific layers from each "module" in the FNN architecture were chosen for the analysis. Why were these chosen, e.g., why only L1 and L13 from the encoder? Could the authors not analyse the evolution of representations through the layers of the network? This goes back to my point about quantifying their results and showing something concrete through their analyses and figures, e.g., if separation by stimuli increases through layers (and quantifying separation through clustering metrics, for example).
* It's important to note that there are different kinds of "foundation models" for neuroscience, and not all of them have to do with predicting neural activity from stimuli. There are several other neural decoding and joint encoding + decoding "foundation model" approaches, e.g.:
  * Decoding, spikes: NDT-2 (https://openreview.net/forum?id=CBBtMnlTGq), POYO (https://openreview.net/forum?id=sw2Y0sirtM), POSSM (https://openreview.net/forum?id=1i4wNFgHDd, incorporates recurrence – important as discussed in this work), NDT-3 (https://openreview.net/forum?id=utXSSdD9mt)
  * Decoding, Calcium imaging, transformer-based: POYO+ (https://openreview.net/forum?id=IuU0wcO0mo)
  * Encoding and decoding, spikes: NEDS (https://openreview.net/forum?id=vOdz3zhSCj)

  There is no discussion on these complementary or related approaches, and I think it would be important to provide an overview of the field and clarify what kind of models the focus is on here.

**Questions:**

* It is unclear to me whether a digital-twin–style neuro-foundation model _must_ itself have computations/representations that resemble the brain – wouldn't strong predictive machine, even if not brain-like, still be useful in the experimental loop? I feel this point lacks justification and could be motivated better.
* On a related note, is it very surprising that these models don't resemble the brain much? They weren't trained on interventional data or with specific architectural details, constraints, or bottlenecks to force their representations to be similar to the brain's. Why then would we expect them to be similar? I would further encourage quantitative analyses as mentioned in my listed weaknesses to substantiate points such as recurrence bringing about better bio-plausibility.
* How were the specific layers analysed selected? Could the authors not look at all layers and how representations/computations evolve through them?
* Could the authors introduce better quantitative comparisons between neural data and models, rather than just qualitative comparisons, such as through CCA, DSA, etc.?
* Why are there no results/figures on encoding manifolds for the neural data itself? I thought, based on the introduction, that a key point was to compare the FNN model's neurons to biological ones on the basis of encoding properties as well?
* Why was the input specification to the FNN model modified, as described in Section A.3? Would the results not be affected by using fewer frames overall, and was there any attempt to finetune the FNN model or provide longer sequences as inputs?

---

> ### Author Response · Authors · 2025-11-22
>
> We thank the reviewer for the detailed and insightful comments. We appreciate the recognition of the importance of our research topic, and the positive assessment of our conceptual approach and of the tubularity metric that we developed. We have worked extensively to address the major concerns regarding __clarity, motivation, and quantitative results__ as reported in our __General Response__ above. In particular, we agree that the previous version was confusing, and are hopeful that the reviewer will find the new version more clear and with a much-improved presentation.
>
> __Specific Questions__
> - __Necessity of brain-like internal representations for digital-twins:__ We agree that a high-fidelity predictive model is useful (e.g., for closed-loop stimulus optimization). Our claim is narrower: when the goal is a digital twin, i.e., a model that supports interventional, counterfactual, and mechanism-level reasoning, then the internal computations do matter. This is due to two main reasons:
> 	1. As we note in the Introduction, a non-mechanistic predictor can match outputs yet fail under out-of-distribution or perturbed regimes that experiments routinely induce (optogenetic silencing, pharmacology, state changes). Our manifold analyses are precisely aimed at this “look inside” requirement: decoding/encoding manifolds and temporal trajectory geometry probe whether the latent state space supports the same separations and dynamics biology uses.
> 	2. If the internal population geometry is misaligned, hypotheses about which subnetworks or computations drive an effect become unreliable. We show concrete mismatches throughout the Results: e.g., the encoder’s padding-driven “intensity arm” and lack of stimulus-dependent temporal structure that would mislead an intervention targeted at “early” processing.
>
> So, we don’t claim that every useful experimental tool must be brain-like; we argue that for digital-twin use-cases, internal alignment is instrumental, and our framework provides the diagnostics to assess it (see also Discussion).
>
> -__Expectations of biological similarity and quantitative evidence for recurrence:__ We don’t mean to suggest that our findings are surprising; rather, our goal is to show __where__ and __how__ current models diverge from biology, and to __quantify__ those differences. At the same time, it’s reasonable to expect some degree of similarity, because:
> 	1. The FNN is trained directly to predict neural activity, not just task labels. Prior work shows that such supervision can lead to unit-level selectivities and RSA-style similarities to biological neurons (see Intro; citations therein). Our results reproduce some of these effects (e.g., orientation/direction selectivity distributions at later stages; cf. Supplemental Figure 16) but also reveal population-level mismatches that standard metrics overlook.
> 	2. Even without interventional data, architectural inductive biases (e.g., convolution, limited temporal kernels) and the predictive objective can naturally promote partial alignment. Our contribution is to measure the degree and location of that alignment rather than to assume it.
>
> Crucially, our goal isn’t to claim that the current FNN should have been brain-like; rather, we want to highlight where and how its internal representations differ in ways that matter for digital-twin applications, and to show with data where biological plausibility starts to appear (recurrent module) and where it falls off (in the encoder and readout stages).
> - __Layer Selection:__ To better cover the full encoder, we have added quantitative results and manifold visualizations for intermediate layers to the Appendix. We restricted our analysis to layers following activation functions to ensure we sample the non-linear feature space of the model. This was done so we could echo the procedure the procedure used in the analysis of neuroscience data; see Dyballa et al. (2024a).
> - __Related Foundation Models:__ We have added the complementary approaches you listed (NDT-2, POYO, NEDS, etc.) to the __Introduction (Prior work)__ to better contextualize our work. Thank you!
> - __Biological Encoding Manifolds:__ We agree that having both versions of the manifolds facilitates comparisons and clarifies the figures. Thus we have recreated the biological encoding manifolds from the data used in Dyballa et al. (2024a) (available online) and added them to __Figure 4__ to allow for direct comparison with the FNN manifolds.
> - __Input Specification:__ We adhere to all FNN specifications. The input sequence length was chosen solely to match the duration of the available biological ground truth data. Since the FNN is trained to predict neural activity iteratively (step-by-step), using the matched sequence length ensures a valid comparison of dynamics without altering the model's fundamental behavior (Section A.3).
>
> We thank you again for your review. Please do not hesitate to reach out in case you have any remaining questions or concerns.

---

> > ### Comment · Reviewer_MUj9 · 2025-11-22
> >
> > Thank you for your response. Based on the authors comments and a quick read, it seems that the paper has changed significantly with these updates. While I appreciate the effort that the authors have put into these updates, it seems to me that the paper will need to be reviewed again entirely given the scale of the changes, which I'm not sure is feasible in the just the discussion phase. I will still try my best to provide comments, but I am not sure I can change my decision strongly just through the discussion phase. Some preliminary comments, however:
> > * In Table 2 you only report the average of all metrics. What about the values of each metric? Furthermore, metrics like DSA need to be understood relative to a control, not just interpreted as one absolute comparison. Maybe a chance similarity  could be shown with completely random or time-shuffled inputs? Also, DSA values are a distance metric, so the lower the DSA, the more similar. Did you take care of this while averaging? I ask because the other metrics like $R^2$ are the opposite.
> > * Quick clarification: so there was no finetuning of the FNN model? I don't see the answer to this (explicitly) in the response to my question.
> > * Thank you for all the other responses, I can understand and the authors' comments on the necessity of brain-like representations with these models.

---

> > > ### Author Response · Authors · 2025-11-22
> > >
> > > Thank you for the quick response. To clarify:
> > > - Full metric values are in the appendix.
> > > - DSA is inverted already. Thanks for the clarification on the DSA baseline, we will look into this and implement a reasonable baseline.
> > > - Yes there was no finetuning of FNN. We apologize for not concretely responding to this. Since our gratings and flow stimuli are similar to training stimuli and the original FNN papers claims generalization to new stimuli, we believe the current FNN behaves adequately on these stimuli after our stimulus adequacy investigations.

---

> > > > ### Author Response · Authors · 2025-11-23
> > > >
> > > > Finally, while we indeed made significant organizational improvements to address clarity concerns (e.g., rewriting sections for better exposition and reorganizing figures), __the main technical contributions and results remain the same. Our goal was to make the paper easier to follow, not to introduce entirely new content__.
> > > >
> > > > We fully understand that your time is limited, but we would greatly appreciate any updated impressions you can share, especially regarding whether the revisions address your earlier clarity concerns.
> > > > Please let us know if any specific parts remain unclear — we’d be happy to clarify during the discussion period.

---

> > > > > ### Author Response · Authors · 2025-11-26
> > > > >
> > > > > Thanks again for your note on the DSA scores, we have now added Z scores to a time-shuffled baseline as suggested. You can find the full metric scores in Table 5.

---

### Author Response · Authors · 2025-11-22
**General response to all reviewers**

First, we would like to thank all reviewers for their insightful, detailed, and constructive comments. This helped us realize that our key message – how the FNN differed from the mouse –  was obscured by the paper’s original organization. We thus rewrote the paper almost completely, changing the organization and figures. The reviewers’ comments were essential in guiding us on how to do this. Previously, the paper was organized by network layer, and the different manifolds and dynamical analysis were interleaved. We have now flipped this around, so that the decoder, encoder, and dynamics now ground the organization (and the Results subsections), with layers shown next to one another.  Thus the development of representations across layers is now much more clear: a reader can simply look at the rows of results in each figure to “see” the differences across layers. We believe this improved the paper enormously – and again thank the reviewers for their motivating comments.

As noted above, there was significant overlap in the reviews, particularly regarding __Clarity/Presentation__ and __Quantitative Results__.  We address these points jointly here and then provide detailed responses to individual questions for each reviewer.

__1. Presentation and Clarity:__ We have substantially revised the manuscript to address concerns regarding the narrative flow and visual presentation. Key updates include:

- __Restructured Introduction:__ We rewrote the introduction to clarify the motivation, the requirements for the "digital twin" approach, and the central narrative (the “story”) of the paper. We emphasize how the input/output map by itself is limited in its explanatory power, and complement it with the “inverse” map and dynamics. Handling each of these three representations, in turn, is the new organization of the Results.  Foundationally,  as we note: “such issues are classical in modeling: control theory teaches us that, without a perfect model, one must “look inside the box” to achieve identifiability (cf. Åström (2012)). We seek to do just this on the Foundation Neural Network (FNN) (Wang et al., 2025). Without this, we cannot guarantee correct, robust, and generalizable behavior, especially on out-of-distribution data, to confidently build hypotheses about the brain using the FNN.” We expand on the interaction between these three representations throughout the revised paper.
- __Enhanced Figures:__ We updated all central figures to improve resolution, readability, and conceptual flow (see specifically the new Figure 1, and the reorganized Figures 3, 4, and 5). We add a new Fig. 2 to illustrate the encoding manifold computation. Again, the reviewers’ comments were extremely helpful in motivating and guiding this reorganization.
- __Manifold Comparisons:__ We restructured the figures showing the resulting manifolds to facilitate direct visual comparison across different model layers.
- __Layer Nomenclature:__ We clarified our layer naming convention. We now count only convolutional layers (excluding pooling operations) to more accurately reflect the network depth and ensure our analysis covers the encoder density adequately.

__2. Quantitative Metrics:__ To address concerns regarding the comparison to quantitative results, we have:
- __Computed Alignment Metrics:__ As suggested by the reviewers, we calculated Representational Similarity Analysis (RSA), Canonical Correlation Analysis (CCA), Linear Predictivity (LP), and Dynamical Similarity Analysis (DSA) scores. They are summarized in Table 2 in the main text and are listed in detail in the Appendix.
- __Additional Results:__ These new results are now included in __Section 3.4__ and in the __Appendix__.

---

### Author Response · Authors · 2025-12-02
**Summary of Reviews and Rebuttal**

Due to the extraordinary circumstances this year, we decided to post a summarizing comment of the rebuttal period. We hope this gives a quick overview of the rebuttal, especially for the new AC who will likely have to manage a heavy workload.

We were fortunate to have very diligent reviewers and would like to thank everyone. The reviewer's concerns were constructive and consistent, and we successfully addressed most of them in the new version of the paper. Specifically, concerns can be grouped into __Presentation/Clarity__, __Motivation__, __Quantitative Evaluation__, __Tubularity__, and __Miscellaneous__. We now describe our changes for every category:

__Presentation/Clarity__: We have restructured the __results section__ to improve reading flow. We revised all __figures__, substantially improving figure quality and __enabling direct comparison of manifolds across layers__. We note that, while this introduced significant changes to the paper, our message and results remained unchanged.

__Motivation__: We have rewritten the __introduction__ to address the reviewers' concerns successfully.

__Quantitative Evaluation__: We have included the suggested metrics from the literature (__RSA, CCA, LP, DSA__), highlighting that our method produces results that __align with these metrics__ and also __uncovers model characteristics not captured by them__.

__Tubularity__: We have added __clarification__ and __theoretical grounding__ for our tubularity metrics and clarified that our goal with this metric is not to introduce a new gold standard of brain alignment but to measure one specific characteristic, supplementing our primary analysis of the manifolds

__Miscellaneous__: We have addressed several minor concerns about missing details in the methods appendix, regarding __FNN architecture, input videos, hyperparameters, sampling procedure__, and more. We have added additional results from __intermediate FNN layers__ and some __sampling tests__ to the appendix to validate our results.

These changes, informed by the reviewers' valuable feedback, have substantially improved the paper. We hope this will help evaluate our submission.

---

### Meta-Review · Area_Chair_5ybj · 2026-01-15

**Summary:**

The paper develops a new tubularity metric for neural activity and uses it and other computational neuroscience methods to analyze foundation model of neural activity. Looking at a foundation model of neural activity in mouse V1 and retina, they find that the models recurrent module aligns best with biology while the feedforward model is less so.

The initial scores were mixed with four rejects (three strong) and one weak accept. The reviewers were largely concerned with the lack of theoretical grounding of the tubularity metric, the poor quality of the presentation (both text and figures), poor motivation and insufficient evaluation. In response, the authors significantly improved the presentation and introduced new analyses. However, a major outstanding concern regarding the soundness of the tubularity metric remains. For this reason, the reviewers opinion of the paper was largely unchanged for the 3 reviewers who responded and the other two are also unlikely to have revised their opinion significantly enough to impact acceptance.

**Reviewer Concerns:**

The initial scores were mixed with four rejects (three strong) and one weak accept. The reviewers were largely concerned with the lack of theoretical grounding of the tubularity metric, the poor quality of the presentation (both text and figures), poor motivation and insufficient evaluation. In response, the authors significantly improved the presentation and introduced new analyses. However, a major outstanding concern regarding the soundness of the tubularity metric remains. For this reason, the reviewers opinion of the paper was largely unchanged for the 3 reviewers who responded and the other two are also unlikely to have revised their opinion significantly enough to impact acceptance.

**Reviewer Scores:**

However, a major outstanding concern regarding the soundness of the tubularity metric remains. For this reason, the reviewers opinion of the paper was largely unchanged for the 3 reviewers who responded and the other two are also unlikely to have revised their opinion significantly enough to impact acceptance.

---

### Decision · Program_Chairs · 2026-01-26

Reject